# From Sequential to Parallel: Reformulating Dynamic Programming as GPU Kernels for Large-Scale Stochastic Combinatorial Optimization

**Jingyi Zhao[1], Linxin Yang[1,2], Haohua Zhang[4], Qile He[2], Tian Ding\*[1,3,4]**
1. Shenzhen Research Institute of Big Data, Shenzhen, China
2. School of Data Science, The Chinese University of Hongkong, Shenzhen, China
3. Shenzhen International Center for Industrial and Applied Mathematics, Shenzhen, China
4. AutoKernel, Shenzhen, China

## Abstract

A major bottleneck in scenario-based Sample Average Approximation (SAA) for stochastic programming (SP) is the cost of solving an exact second-stage problem for every scenario, especially when each scenario contains an NP-hard combinatorial structure. This has led much of the SP literature to restrict the second stage to linear or simplified models. We develop a GPU-based framework that makes structured integer recourse operators tractable at scale. The key innovation is a set of hardware-aware, scenario-batched GPU kernels that expose parallelism across scenarios, dynamic-programming (DP) layers, and route or action options, enabling Bellman updates to be executed in a single pass over more than $10^6$ realizations. We evaluate the approach in two representative SP settings: a vectorized split operator for stochastic vehicle routing and a DP for inventory reinsertion. Implementation scales nearly linearly in the number of scenarios and achieves a one-two to four–five orders of magnitude speedup, allowing far larger scenario sets and reliably stronger first-stage decisions. The computational leverage directly improves decision quality: much larger scenario sets and many more first-stage candidates can be evaluated within fixed time budgets, consistently yielding stronger SAA solutions. Our results show that structured integer recourse operators are tractable at scales previously considered impossible, providing a practical path to large-scale, realistic stochastic discrete optimization.

## 1 Introduction

A central challenge in scenario-based stochastic programming (SP) is the computational burden of solving an exact second-stage problem for every scenario, especially when each scenario embeds an NP-hard combinatorial structure. This difficulty has led much of the SP literature to restrict the second stage to linear programs or simplified models (Birge & Louveaux, 1997; Shapiro et al., 2021), sacrificing realism and often degrading first-stage decision quality. Our goal is to make full-fidelity, integer second-stage models computationally viable at scale, without relying on surrogate learning models or structural relaxations.

A key observation is that once a first-stage decision is fixed, second-stage recourse evaluations across scenarios are independent. Many of these evaluations rely on dynamic programming (DP), a foundational paradigm in optimization and control (Bertsekas, 2012) and a core ingredient in both exact and heuristic combinatorial optimization. DP underlies classical algorithms such as the Held–Karp procedure for TSP (Held & Karp, 1971), pseudo-polynomial knapsack solvers (Shapiro, 1968), and shortest path algorithms (Bellman, 1958; Dijkstra, 1959). It appears as a subroutine in vehicle and inventory routing—via split operators (Prins, 2004; Vidal, 2016) and resource-constrained shortest-path pricing (Feillet et al., 2004; Irnich & Desaulniers, 2005)—and in large-scale integer programs (Barnhart et al., 1998). DP also plays a central role in metaheuristics (Vidal et al., 2012; Zhao et al.,

2022) and multistage stochastic optimization (Powell, 2007; Bertsekas & Tsitsiklis, 1995; Staddon, 2020).

Despite this ubiquity, most DP implementations are inherently sequential, obscuring the substantial parallelism available across scenarios and within DP transitions. While recent progress in GPU acceleration has benefited continuous optimization (Schubiger, 2019; Lu & Yang, 2023; Bishop et al., 2024; Liu et al., 2024), no comparable framework exists for discrete, combinatorial, scenario-based problems.

We address this gap by developing a GPU-based framework that executes second-stage dynamic programs in a *scenario-batched* and *multidimensional* manner. The key novelty lies in custom GPU kernels that expose parallelism simultaneously across scenarios, DP layers, and route or action options. These kernels implement Bellman updates using warp-/block-level reductions with numerically safe masking, enabling a single GPU pass to evaluate over $10^6$ uncertainty realizations—far beyond what existing SP approaches can handle.

We demonstrate the approach on two benchmark problems that capture core challenges in stochastic routing and inventory optimization: (i) a vectorized split operator for capacitated vehicle routing with stochastic demand (CVRPSD), and (ii) an inventory reinsertion DP under an order-up-to policy. Across benchmarks, our implementation scales nearly linearly with the number of scenarios and achieves substantial speedups over CPU baselines—from one–two orders of magnitude for CVRPSD to four–five orders for DSIRP. Additional experiments show that these gains translate directly into better decision quality: the ability to evaluate far more scenarios and far more first-stage candidates within the same time budget consistently yields stronger solutions. Our implementation is available at `https://github.com/Jingyi-poly/2-stage-IRP-GPU/tree/CVRPSD-split-GPU` and `https://github.com/Jingyi-poly/2-stage-IRP-GPU/tree/OUIRP-GPU`.

In summary, our main contributions are as follows.

1. We expose and formalize the multidimensional parallelism inherent in second-stage dynamic programs for stochastic combinatorial optimization, and demonstrate that this structure enables scenario-batched execution at scales previously out of reach, surpassing $10^6$ scenarios.

2. We design hardware-aware GPU kernels that exploit concurrency across scenarios, DP layers, and route/action options. These kernels implement efficient Bellman reductions with numerically safe masking, enabling high-throughput evaluation of massive scenario batches.

3. We present extensive empirical evidence, including comparisons against the extensive-form MILP solved by Gurobi for CVRPSD and against the state-of-the-art DSIRP algorithm (Coelho et al., 2012). Our method remains tractable for far larger scenario sets and frequently obtains higher-quality solutions.

4. We demonstrate that GPU-accelerated second-stage evaluation substantially strengthens metaheuristic search: orders-of-magnitude more first-stage candidates and far larger scenario sets can be explored within the same time budget. In SAA, solution quality improves reliably as the number of scenarios increases, making our computational breakthrough directly valuable for obtaining high-quality stochastic solutions.

5. We provide a detailed GPU-memory study showing that realistic problem instances with up to $10^6$ scenarios fit within a standard 11GB GPU, confirming that memory is not the limiting factor at scale. We also outline a general recipe for converting DP subroutines into high-throughput GPU primitives.

## 2 GENERIC DYNAMIC PROGRAMMING FRAMEWORK

**Preliminary Knowledge.** We consider a finite-horizon dynamic program over stages $t = 1, \ldots, T$, starting from an initial state $s_1 \in \mathcal{S}_1$. At each stage $t$, the system is in a state $s_t \in \mathcal{S}_t$, an action $a_t \in \mathcal{A}_t(s_t)$ is chosen, and the system moves to $s_{t+1}$ either deterministically via $g_t(s_t, a_t)$ or stochastically according to $P_t(s_{t+1} \mid s_t, a_t; \omega)$. Each transition incurs a stage cost $c_t(s_t, a_t; \omega)$ under scenario $\omega$.

Let $J_t^\omega(s)$ denote the minimum cumulative cost to reach $s \in \mathcal{S}_t$ at stage $t$. The recursion initializes as $J_1^\omega(s_1) = 0$ and evolves by

$$J_{t+1}^\omega(s') = \min_{\substack{s \in \mathcal{S}_t,\, a \in \mathcal{A}_t(s) \\ g_t(s,a)=s'}} \{J_t^\omega(s) + c_t^\omega(s,a)\}, \quad \forall s' \in \mathcal{S}_{t+1}. \tag{1}$$

The objective is to minimize the expected terminal cost $\mathbb{E}_\omega\left[ \min_{s \in \mathcal{S}_T} J_T^\omega(s)\right]$.

## 2.1 TRANSITION-BASED FORMULATION.

To enable GPU-friendly computation, we express the recursion in terms of state-to-state transition costs. For each stage $t$ and scenario $\omega$, define

$$A_t^\omega(s,s') := \inf_{\substack{a \in \mathcal{A}_t(s) \\ g_t(s,a)=s'}} c_t^\omega(s,a), \qquad A_t^\omega(s,s') = +\infty \text{ if no feasible transition exists.}$$

The forward recursion then becomes

$$J_{t+1}^\omega(s') = \min_{s \in \mathcal{S}_t}\{J_t^\omega(s) + A_t^\omega(s,s')\}.$$

This formulation results in a stage-wise min–plus update that can be evaluated independently for each stage, making it well aligned with batched GPU execution.

## 2.2 MIN-PLUS MATRIX FORMULATION.

Let $\mathcal{S}_t = \{1,\ldots,m_t\}$ index the state space. Define:

$$A_t^\omega(i,j) = A_t^\omega(s=i,\, s'=j), \qquad J_t^\omega \in \mathbb{R}^{m_t}.$$

Then, the Bellman update becomes a matrix-vector product in the $(\min,+)$ semiring:

$$J_{t+1}^\omega = (A_t^\omega)^\top \otimes J_t^\omega := \left[\min_i\{A_t^\omega(i,j) + J_t^\omega(i)\}\right]_{j=1}^{m_{t+1}}. \tag{2}$$

This min-plus formulation enables efficient GPU implementation via tensor broadcasting and dimension-wise minimization, with infeasible transitions masked via $+\infty$. For variable-sized state spaces, padding and masking ensure regular tensor shapes for parallel execution. A demonstrative DP example (A.2), together with the full formulations of the following two applications, CVRPSD split (A.3) and DSIRP reinsertion (A.4), is provided in the Appendix.

## 2.3 INSTANTIATION A: SPLIT DP ON A GIANT TOUR IN THE VEHICLE ROUTING PROBLEM WITH STOCHASTIC DEMAND.

**Problem Motivation.** In vehicle routing postprocessing, a common task is to split a "giant tour" $\sigma = [\sigma_1,\ldots,\sigma_n]$ into capacity-feasible routes. Given demands $q_{\sigma_k}^\omega$ under scenario $\omega$ and vehicle capacity $Q$, define state $i$ as having served customers $\sigma_1$ to $\sigma_i$. An action $p < i$ ends the previous route at $p$, starting a new one from $p+1$ to $i$. The departure depot is denoted by $0$, and the destination depot by $n+1$. The DP explores all possible cut points $p < i$ that define where to start a new route, and accumulates the minimal total travel cost for serving customers up to $i$. (see the Figure 9 in A.3 for better understanding).

**Forward DP Recursion and Matrix Form.** The cost of serving subroute $[\sigma_{p+1},\ldots,\sigma_i]$ is

$$W^\omega(p,i) = c_{0,\sigma_{p+1}} + \sum_{k=p+1}^{i-1} c_{\sigma_k,\sigma_{k+1}} + c_{\sigma_i,n+1},$$

which is feasible only if $\sum_{k=p+1}^i q_{\sigma_k}^\omega \leq Q$. We define masked transition entries as

$$A^\omega(p,i) = \begin{cases} W^\omega(p,i), & \text{if } p < i \text{ and } \sum_{k=p+1}^i q_{\sigma_k}^\omega \leq Q, \\ +\infty, & \text{if } p < i \text{ and } \sum_{k=p+1}^i q_{\sigma_k}^\omega > Q \quad \text{(capacity violated)}, \\ +\infty, & \text{if } p \geq i \quad (\text{ not applicable}). \end{cases}$$

Let $J^\omega(0) = 0$ and $J^\omega(i)$ be the optimal cost to reach state $i$. The forward-DP update is then

$$J^\omega(i) \;=\; \min_{p<i}\{J^\omega(p) + A^\omega(p,i)\}, \qquad i = 1, \ldots, n,$$

which is equivalently expressed as the masked min–plus reduction

$$J^\omega(i) \;=\; \big(A^\omega(\cdot,i)\big)^\top \otimes J^\omega(0{:}i{-}1), \tag{3}$$

where $\otimes$ denotes the $(\min, +)$ semiring product and $A^\omega(\cdot, i)$ is the $i$-th column with all infeasible or undefined entries masked by $+\infty$ (see Appendix A.3 for a numerical toy example).

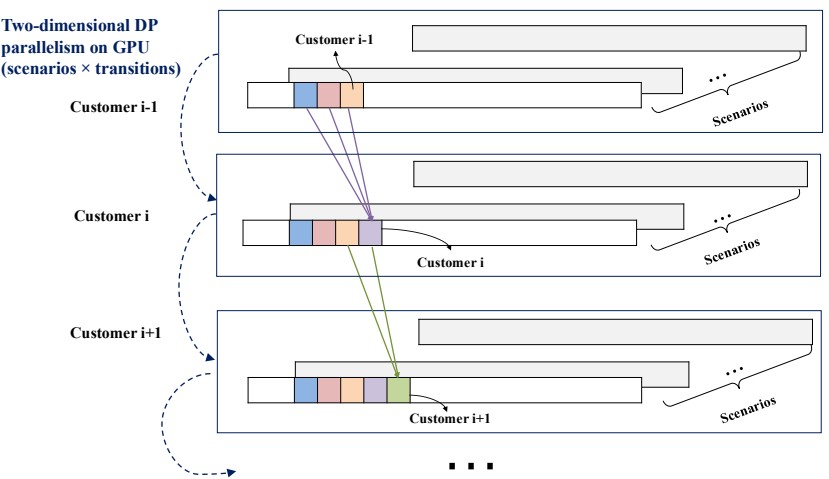

Figure 1: 2D DP parallelism on GPU (scenarios $\times$ predecessors). Each row corresponds to a destination state $i$, each block within a row represents a scenario $\omega$, and colored bars indicate feasible predecessors $p < i$. For each $(i, \omega)$ pair, all predecessors are expanded in parallel to form the set $\{J^\omega(p) + A^\omega(p,i) : p < i\}$, followed by a column-wise min-reduction over $p$ that yields $J^\omega(i)$.

**2D Parallelism on GPU.** From equation 3, the computation at a fixed destination state $i$ factorizes over the Cartesian product $\Omega \times \{p : p < i\}$. We therefore exploit 2D GPU parallelism across **scenarios** $\omega$ and **predecessors** $p < i$. Each thread computes one pair $(\omega, p)$ by loading $A^\omega(p, i)$ and the partial cost $J^\omega(p)$, forming $J^\omega(p) + A^\omega(p, i)$. A warp-/block-level *min* reduction across $p$ then yields $J^\omega(i)$ for that scenario. Launching such kernels for all scenarios in parallel computes the masked min–plus reduction of $A^\omega(\cdot, i)$ against $J^\omega(0{:}i{-}1)$. As illustrated in Figure 1, this structure maps naturally to GPUs: scenarios $\omega$ are parallelized across columns, predecessors $p$ are reduced within blocks, and rows (states $i$) advance independently along the DP frontier.

## 2.4 Instantiation B: Forward Inventory Reinsertion DP in Dynamic Stochastic Inventory Routing Problems.

**Problem Motivation.** In the stochastic inventory routing problem, delivery schedules are often determined at an aggregate level and then refined through local reinsertion moves: a customer $i$ that risks a stockout is reconsidered, and new visits are inserted into existing routes or additional trips are scheduled. The key challenge is that reinsertion must balance two competing effects: (i) earlier deliveries create higher holding cost and may cause inefficiency in vehicle loading, (ii) postponing deliveries increases the probability of future stockouts under adverse demand realizations. The goal is to decide, for each customer, *when* to replenish under uncertain demand so as to minimize expected routing, holding, and stockout costs. DP provides a natural way to resolve this trade-off, as it captures the temporal coupling of inventory states and demand uncertainty.

The decision process follows a two-stage stochastic optimization framework. In the first stage, a delivery and routing plan is established for Day 1. Specifically, the model determines which customers to replenish and how much to deliver, subject to vehicle capacity and routing constraints. These decisions are made prior to the realization of demand and are identical across all demand

scenarios. Once the demand over the entire planning horizon (Days 1 to $H$) is realized, second-stage decisions are made adaptively from Day 2 onward. These include scenario-dependent routing and replenishment actions that respond to the realized demand in each scenario.

The objective is to *minimize the total expected cost*, which consists of two components: (1) First-stage costs including the Day 1 routing cost and the supplier's inventory holding cost; and (2) Second-stage costs, which vary across scenarios and include: inventory holding costs and stock-out penalties at customers at the end of Day 1; routing and delivery costs from Day 2 to $H$; and inventory holding and stock-out penalties from Day 2 to $H$.

The DP operator used to solve this problem follows the formulation in (Zhao et al., 2025), which focuses on a single customer $i$ over a planning horizon $t = 1, \ldots, H$, starting with initial inventory $I_i^0$ and capacity $U_i$. The customer's demand is uncertain and modeled by a finite set of scenarios $\Omega$, where $d_i^{t,\omega}$ denotes demand on day $t$ under scenario $\omega$. For each customer, the decision at each day consists of:

- whether customer $i$ is visited at time $t$,
- the delivery quantity $q_i^t$, which can only take two values: $q_i^t \in \{0, U_i - I_i^{t-1}\}$ that is, either no delivery or replenishment up to full capacity.
- and which vehicle route is chosen to accommodate this visit.

These decisions are scenario-independent, i.e., the same schedule applies across all $\omega \in \Omega$ while inventory evolution is scenario-dependent. The state variable is the end-of-day inventory $I_i^{t,\omega}$, updated as

$$I_i^{t,\omega} = \max\{0, \ I_i^{t-1,\omega} + q_i^t - d_i^{t,\omega}\}, \quad \forall t, \ \omega.$$

Here the DP systematically evaluates both replenishment options (no delivery vs full OU delivery), propagating inventory states forward in time and accumulating costs. See Appendix A.4 for a numerical toy example.

**Forward DP Recursion and Matrix Form.** At each day $t$, customer $i$ either receives no delivery ($q_i^t = 0$) or is replenished up to capacity ($q_i^t = U_i - I_i^{t-1,\omega}$). The per-stage cost for scenario $\omega$ consists of two components: (1) routing and detour costs associated with sending $q_i^t$, denoted $F_t(q_i^t)$; and (2) customer-side inventory holding and stock-out penalties $h_i^t(I_i^{t,\omega})$, evaluated at the end-of-day inventory $I_i^{t,\omega}$.

Let $C_i^t(I_i^{t,\omega})$ denote the minimum expected cumulative cost up to day $t$ for customer $i$ under scenario $\omega$, given that the day-$t$ starting inventory is $I_i^{t,\omega}$. By construction, $C_i^t(\cdot)$ is a piecewise linear function of the inventory state. The forward recursion is

$$C_i^{t+1}(I_i^{t+1,\omega}) = \min_{q_i^t \in \{0, U_i - I_i^{t,\omega}\}} \left\{ C_i^t(I_i^{t,\omega}) + F_t(q_i^t) + h_i^t(I_i^{t+1,\omega}) \right\},$$

where the inventory state evolves as $I_i^{t+1,\omega} = \max\{0, \ I_i^{t,\omega} + q_i^t - d_i^{t,\omega}\}$. The recursion starts from the initial inventory before day 1: $C_i^0(I_i^{0,\omega}) = 0$ with $I_i^{0,\omega} = I_i^0$.

To enable GPU-friendly computation, we collapse the action space into a state-to-state transition matrix:

$$A_i^{t,\omega}(I, J) := \min_{\substack{q \in \{0, U_i - I\} \\ \max\{0, I + q - d_i^{t,\omega}\} = J}} \left\{ F_t(q) + h_i^t(J) \right\}, \quad +\infty \text{ if no feasible } q \text{ leads from } I \text{ to } J.$$

In practice, evaluating each entry $A_i^{t,\omega}(I, J)$ may itself require enumerating a finite set of candidate route options (e.g., alternative reinsertion choices), in which case

$$A_i^{t,\omega}(I, J) = \min_{r \in \mathcal{K}} A_i^{t,\omega}(I, J; r).$$

Rows correspond to today's starting inventory $I$, columns to tomorrow's inventory $J$, and each entry stores the minimal cost of transitioning from $I$ to $J$.

Define the state space

$$\mathcal{S}_i^t := \{I \in \mathbb{Z}_{\geq 0} : 0 \leq I \leq U_i\},$$

and collect $C_i^t(I)$ over $I \in \mathcal{S}_i^t$ as a vector $J_i^t \in \mathbb{R}^{|\mathcal{S}_i^t|}$. The forward recursion then becomes a masked min–plus matrix–vector product:

$$J_i^{t+1} = (A_i^{t,\omega})^\top \otimes J_i^t := \left[ \min_{I \in \mathcal{S}_i^t} \left\{ A_i^{t,\omega}(I, J) + J_i^t(I) \right\} \right]_{J \in \mathcal{S}_i^{t+1}}. \tag{4}$$

**3D Parallelism on GPU.** From the matrix form, the computation at stage $t$ factorizes over the Cartesian product $\Omega \times \{(I \to J)\} \times \mathcal{R}$, where $\mathcal{R}$ denotes the set of candidate route options for each transition. We therefore exploit 3D GPU parallelism across **scenarios** $\omega$, **state transitions** $I \to J$, and **route options** $r \in \mathcal{R}$. Each thread computes one tuple $(\omega, I \to J, r)$ by loading $A_i^{t,\omega}(I, J; r)$ and the partial cost $J_i^t(I)$, forming $J_i^t(I) + A_i^{t,\omega}(I, J; r)$. A warp-/block-level *min* reduction is first performed across route options $r$, then across predecessor states $I$, yielding $J_i^{t+1}(J)$ for each scenario $\omega$. Launching such kernels for all $\omega$ in parallel realizes the batched column-wise min–plus updates of the recursion, while also vectorizing over alternative delivery routes.

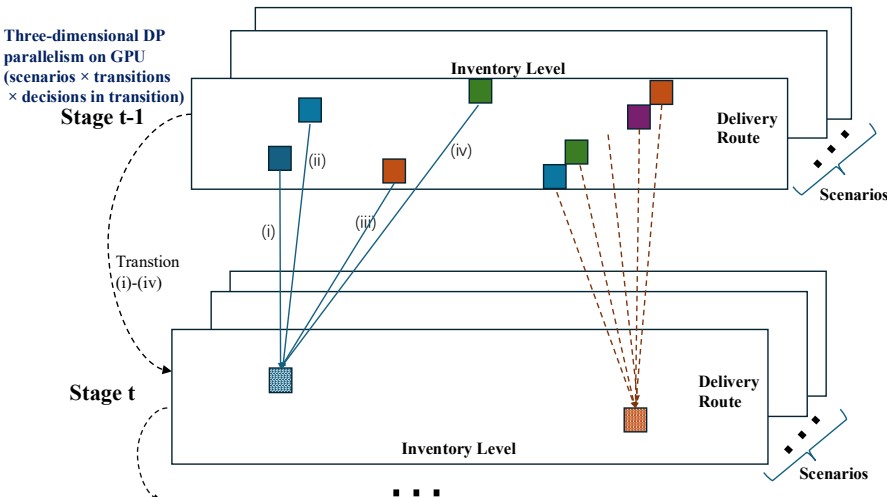

Figure 2: 3D DP parallelism on GPU (scenarios $\times$ transitions $\times$ route options). Each layer corresponds to a stage $t$, with nodes representing end-of-day inventory levels $I$. Colored edges denote feasible transitions $I \to J$ under scenario-specific demands $d^{t,\omega}$. For each tuple $(\omega, I \to J, r)$, threads evaluate the cost contribution $J_i^t(I) + A_i^{t,\omega}(I, J; r)$, combining routing overhead with holding and stockout penalties. A two-level reduction (first across route options $r$, then across predecessor states $I$) yields $J_i^{t+1}(J)$ per scenario. The figure highlights how GPU parallelism spans scenarios, transitions, and route options, turning the DP recursion into a fully batched min–plus update.

## 3  EXPERIMENTS

In this section, we present numerical experiments demonstrating the effectiveness and practicality of the proposed matrix-form DP framework. Section 3.1 examines the role of large scenario sets in improving solution quality, and Section 3.2 shows the scalability of our approach. Sections 3.3 and 3.4 verify that these findings also hold for our GPU-based implementation, while Section 3.5 evaluates its feasibility in terms of computational resources. Additional comparisons with baseline methods are provided in Appendices A.6–A.7, with Appendix A.8 analyzing the effect of the number of evaluated first-stage solutions.

### 3.1  SCALING THE SCENARIO SIZE IN STOCHASTIC PROGRAMMING.

The theoretical properties of empirical risk minimization (ERM) under mild regularity conditions establish that SAA solutions may suffer from *bias* with small sample sizes but converge *consistently* toward the true optimum as the number of scenarios grows, with an asymptotic $\mathcal{O}(1/\sqrt{m})$ convergence rate (see Appendix A.5 for a formal statement). To examine how these properties manifest in

practice, we conduct experiments on the DSIRP, a representative setting where demand distributions are complex and cannot be adequately captured by simple parametric families. Figure 3 reports the empirical behavior of SAA estimators as the number of scenarios increases.

Specifically, Figures 3a and 3b highlight the bias effect: with only a small number of scenarios, the estimated cost systematically underestimates the true expectation. As the sample size grows, the estimator increases and gradually stabilizes, consistent with the theoretical consistency guarantee. Figures 3c and 3d further illustrate the convergence behavior. As the number of scenarios increases from hundreds to tens of thousands, the SAA estimate approaches the true optimum, and its variance decreases at the predicted $\mathcal{O}(1/\sqrt{m})$ rate. The log-scaled plot shows that improvements continue to accrue even at large sample sizes, underscoring that "more scenarios" consistently yield better estimates rather than reaching a premature plateau.

In general, these findings demonstrate that when the underlying distribution of uncertainty is complex or unknown, as is common in real-world, data-driven settings, restricting the scenario set to only a few hundred or a few thousand is insufficient. Substantially larger scenario sets are needed to reduce bias and improve solution quality. Our GPU-accelerated DP framework makes such large-scale, data-driven stochastic evaluation computationally feasible in practice.

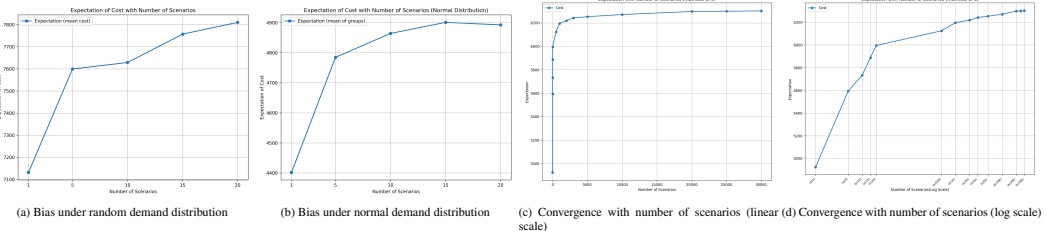

(a) Bias under random demand distribution  (b) Bias under normal demand distribution  (c) Convergence with number of scenarios (linear scale)  (d) Convergence with number of scenarios (log scale)

Figure 3: Empirical behavior of SAA estimators in DSIRP. Top: bias under different demand distributions. Bottom: convergence with increasing scenario size.

## 3.2 SCALABILITY WITH THE NUMBER OF SCENARIOS.

The above results confirm that larger scenario sets are statistically necessary to reduce bias and achieve consistency in stochastic programming. We next examine whether such scaling is computationally feasible. We evaluate the efficiency of our GPU-accelerated DP operators against CPU baselines on two representative tasks: (i) the split operator in CVRPSD and (ii) the reinsertion operator in DSIRP. All implementations were written in C++/CUDA and tested on a machine with an AMD Ryzen 7 9700X CPU (8 cores) and an NVIDIA RTX 2080Ti GPU with 11 GB memory. The CPU baselines include (i) a single-threaded implementation and (ii) a multi-threaded implementation with 8 threads. The GPU implementations exploit the 2D/3D parallelism described in Section 2.

**Left (CVRPSD split DP).** As the number of scenarios increases from $10^4$ to $10^6$, the single-thread CPU runtime rises sharply and reaches minutes at $10^6$ scenarios, while the 8-thread baseline shows only moderate relief before saturating due to synchronization and memory-bandwidth limits. The GPU curve grows nearly linearly and remains in the seconds range even at $10^6$ scenarios, yielding about $80\times$ speedup over single-thread CPU and $20\times$ over the 8-thread baseline at the largest setting.

**Right (DSIRP reinsertion DP).** The effect is far more dramatic. At $2 \times 10^5$ scenarios, the GPU implementation attains roughly $9.3 \times 10^4$ speedup versus the single-thread CPU and $2.26 \times 10^4$ versus the multi-thread CPU (see callouts in the figure). These gains stem from: (i) 3D parallelism (scenarios $\times$ inventory transitions $\times$ route options), (ii) high arithmetic intensity with coalesced memory access, and (iii) warp-/block-level reductions that keep the Bellman minima on-chip.

Across both problems, GPU parallelization shifts the computational frontier for scenario-based evaluation: near-linear scaling in the number of scenarios with order-of-magnitude to five-orders-of-magnitude speedups (problem-dependent). This throughput is what enables the very large, data-driven scenario sets used in our SAA experiments, directly supporting the statistical benefits documented in the previous subsection.

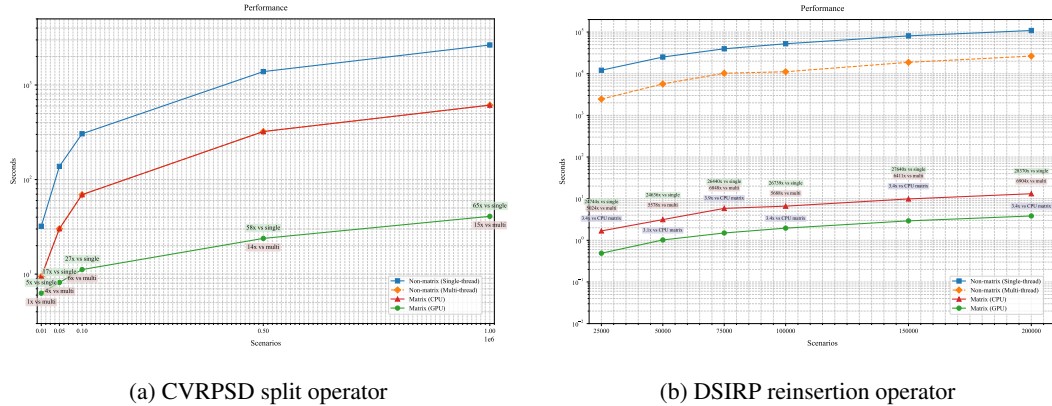

(a) CVRPSD split operator        (b) DSIRP reinsertion operator

Figure 4: Runtime comparisons of CPU and GPU implementations across different scaling regimes. Left: scaling behavior under $10^4$–$10^6$ scenarios for CVRPSD. Right: large-scale evaluation for DSIRP.

### 3.3 IMPACT OF SEARCHING SCENARIO SET SIZE ON DECISION QUALITY

We next examine how the number of evaluated scenarios influences the quality of the first-stage decision. Specifically, we solve the problem under different scenario counts, ranging from 1 to $10^4$ (i.e., $1, 100, 1,000$). For each scenario setting, the obtained first-stage solution is evaluated on a fixed large out-of-sample evaluation set of $10^6$ scenarios. Figure 5 reports the out-of-sample cost achieved by the best observed solution on two CVRPSD instances: x-n128 and x-n105.

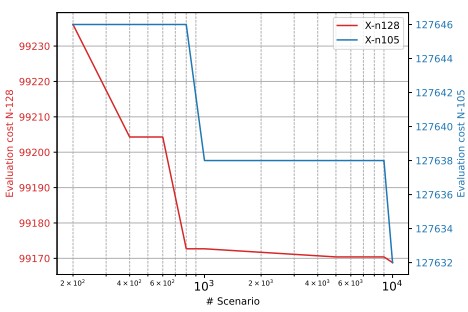

Figure 5: Out-of-sample performance of first-stage solutions obtained with varying observed scenario settings. Larger evaluation set yield more robust and lower-cost solutions.

Figure 6: Quality of the best solution obtained at each time under a fixed time budget. GPU consistently achieves better decisions due to faster evaluation and thus larger effective search effort.

**Results.** When observing only few scenarios (e.g., a single scenario), the resulting first-stage solution is severely biased and performs poorly. Increasing the scale of available scenarios consistently improves robustness, with significant gains observed throughout Figure 5. Such performance confirms the theoretical insight that larger sample sizes reduce estimation bias in sample-average approximation. These results demonstrate that our GPU-based framework, by enabling the evaluation of tens or hundreds of thousands of scenarios within practical runtimes, leads to significantly more reliable first-stage solutions compared to CPU-based methods that are restricted to only a few thousand scenarios.

### 3.4 DECISION QUALITY UNDER FIXED TIME BUDGETS

Finally, we compare the decision quality obtained under identical wall-clock time limits across the three implementations: CPU single-thread, CPU multi-thread, and GPU. Each method is given a fixed runtime budget , during which the split algorithm splits giant tours to obtain first-stage solu-

tions. For fairness, all approaches are evaluated on the same problem instance with $10^4$ available scenarios. Figure 6 reports the best penalized cost obtained within the allowed runtime.

**Results.** With small time budgets, all methods return feasible but suboptimal solutions, yet GPU already provides a noticeable advantage. As the time limit increases, the quality gap widens: GPU produces solutions that are consistently closer to the true optimum, while CPU single-thread stagnates and CPU multi-thread improves only modestly. This matches intuition: faster scenario evaluation allows the GPU to explore many more candidate first-stage tours within the same runtime, thereby improving the probability of discovering high-quality solutions.

These results confirm that beyond scalability, GPU acceleration directly translates into superior decision quality under realistic time constraints, making it particularly valuable in operational settings where decisions must be made quickly.

### 3.5 GPU-USAGE ANALYSIS

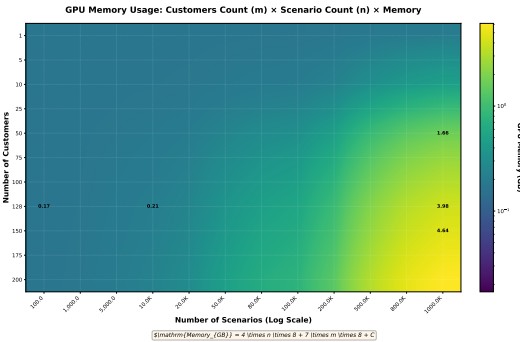

Figure 7: Relationship between GPU utilization and the number of scenarios/customers in CVRPSD instances.

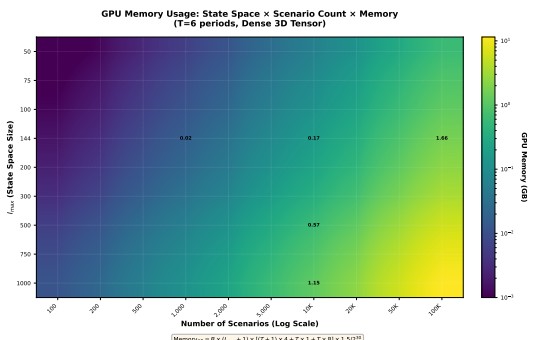

Figure 8: Relationship between GPU utilization and the number of scenarios/$I_{m}ax$ in DSIRP instances.

To assess the viability of our method with respect to GPU memory requirements, we conduct dedicated experiments for both the GPU-based CVRPSD and DSIRP solvers. Figures 7 and 8 summarize the resulting GPU-memory consumption. Results for both CVRPSD and DSIRP show that GPU memory usage grows strictly linearly with the number of scenarios and the state-space size. Even large configurations, such as $10^6$ scenarios for CVRPSD (3.98 GB) or 100,000 scenarios for DSIRP (1.66 GB), can fit easily within common GPU limits, implying that an 11 GB GPU can support several million scenarios. This confirms that GPU memory is not a practical bottleneck; scalability is limited by computation time rather than memory capacity.

## 4 CONCLUSION

We presented a GPU-based framework for executing second-stage dynamic programs in a scenario-batched and multidimensional manner, enabling full-fidelity stochastic combinatorial optimization at scales that were previously impractical. By exploiting parallelism across scenarios, DP layers, and action or routing choices, our kernels can evaluate over a million uncertainty realizations in a single pass. Experiments on a vectorized split operator for stochastic vehicle routing and an inventory reinsertion dynamic program show that the method scales nearly linearly with the number of scenarios and yields substantial speedups over multithreaded CPU baselines, leading directly to stronger first-stage decisions under the same computational budget. Although the framework inherits GPU memory limitations and currently focuses on forward dynamic programs with additive costs, it opens several promising directions. Future work includes handling constrained and risk-aware objectives, designing GPU-native abstractions for irregular dynamic programs, integrating learned components while preserving numerical reliability, and extending the primitives beyond the specific problems studied here. We view this work as an early step toward hardware-aware algorithms for large-scale stochastic discrete optimization.

ACKNOWLEDGMENTS

This work was supported by the Longgang District Special Funds for Science and Technology Innovation under Grant LGKCSDPT2023002.

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

# A APPENDIX

## A.1 RELATED WORKS

DP has long been a foundation of combinatorial optimization and stochastic control Carraway et al. (1989); Chu (2011); Paschos (2014), underpinning exact algorithms for the traveling salesman problem Bellman (1962); Malandraki & Dial (1996), knapsack variants Bertsimas & Demir (2002), and shortest-path computations Ferone et al. (2021); Righini & Salani (2008). In vehicle and inventory routing, DP-based split and labeling procedures are core subroutines in state-of-the-art heuristics and hybrid genetic/metaheuristics Vidal (2016); Cattaruzza et al. (2014), enabling fast evaluation and repair of giant tours and complex neighborhoods. In this sense, DP forms the backbone of both exact and heuristic combinatorial solvers. MILP formulations offer another widely used modeling approach for VRPs Brahimi & Aouam (2016); Song et al. (2018), but solving such models using CPU-based solvers remains computationally intensive, especially for stochastic VRPs with large scenario sets.

Moreover, GPU acceleration has gained traction in optimization and control, with notable developments in GPU-ADMM Schubiger et al. (2020), primal–dual methods for linear and quadratic programming Lu & Yang (2025), and GPU-based interior-point algorithms Gade-Nielsen (2014). These works demonstrate that careful kernel design and memory organization can deliver substantial speedups relative to multithreaded CPU implementations. Researchers have also explored GPU acceleration for routing problems, either through algorithms specifically tailored to GPU architectures or by leveraging high-level GPU libraries. For instance, Boschetti et al. (2017) accelerates q-route–type DP relaxations for VRPs on GPUs, while Maleki et al. (2014) introduces a parallel linear–tropical DP algorithm that circumvents cross-stage dependencies via rank convergence. On the library side, Davis (2019); Yang et al. (2022) implement GraphBLAS for sparse linear algebra on GPUs, and Farrington et al. (2023) uses the GPU-enabled JAX framework to accelerate DP-based stochastic control and routing computations.

Despite these advances, current approaches lack structural generality. GPU-oriented libraries such as JAX require reformulating problems into linear-algebra primitives to match the library's abstraction, while algorithm-specific GPU designs rely on bespoke kernels and custom data layouts, limiting portability across problem classes. To the best of our knowledge, no unified framework currently exists for accelerating DP-based stochastic combinatorial optimization on GPUs. A general-purpose framework would make GPU acceleration reusable across a broad range of DP-based problems, eliminating the need to redesign complex kernels for each new application.

## A.2 Illustrative example for the min-plus matrix-vector Bellman update.

To illustrate the min-plus matrix-vector Bellman update, consider a dynamic programming recursion over three states in stage $t$ and two states in stage $t + 1$.

Let the cost-to-go vector at stage $t$ under scenario $\omega$ be:

$$J_t^\omega = \begin{bmatrix} 0 \\ 1 \\ 3 \end{bmatrix},$$

and the transition cost matrix from stage $t$ to $t + 1$ be:

$$A_t^\omega = \begin{bmatrix} 2 & 5 \\ 1 & +\infty \\ 3 & 0 \end{bmatrix},$$

where $A_t^\omega(i, j)$ denotes the cost of transitioning from state $i$ at stage $t$ to state $j$ at stage $t + 1$. The value $+\infty$ denotes an infeasible transition.

Then, the Bellman update in the $(\min, +)$ semiring becomes:

$$J_{t+1}^\omega = (A_t^\omega)^\top \otimes J_t^\omega,$$

which is computed entry-wise as:

$$J_{t+1}^\omega(1) = \min_i \{A_t^\omega(i, 1) + J_t^\omega(i)\} = \min\{2 + 0,\, 1 + 1,\, 3 + 3\} = \min\{2, 2, 6\} = 2,$$

$$J_{t+1}^\omega(2) = \min_i \{A_t^\omega(i, 2) + J_t^\omega(i)\} = \min\{5 + 0,\, +\infty + 1,\, 0 + 3\} = \min\{5, +\infty, 3\} = 3.$$

Thus, the updated cost-to-go vector at stage $t + 1$ is:

$$J_{t+1}^\omega = \begin{bmatrix} 2 \\ 3 \end{bmatrix}.$$

This computation reflects a forward shortest-path propagation over a layered graph, where the cost of reaching each state in the next stage is determined by minimizing over all incoming transitions from the previous stage.

---

**Algorithm 1** High-level Scenario-Batched GPU Forward DP

---

**Input:** Scenarios $\Omega$, horizon $T$, state sets $\{\mathcal{S}_t\}$, transition-cost generator $\mathcal{A}_t^\omega$, initial cost-to-go $J_1^\omega$
**Output:** Final cost-to-go vectors $\{J_t^\omega\}_{\omega \in \Omega,\, t=1..T}$
 1: Preprocess all scenario tensors (pad, mask infeasible transitions, move to device)
 2: **for** $t = 1$ to $T - 1$ **do**
 3:     Launch GPU kernel over the chosen parallel axes:
       – scenarios $\omega \in \Omega$
       – state transitions $(s \to s') \in \mathcal{S}_t \times \mathcal{S}_{t+1}$
       – optional actions $a \in \mathcal{A}_t(s)$ (if present)
 4:     Each thread:
       load $J_t^\omega(s)$
       load transition cost $A_t^\omega(s, s'; a)$
       compute partial cost $c = J_t^\omega(s) + A_t^\omega(s, s'; a)$
 5:     Within warp/block:
       reduce $\min$ over actions $a$ (if applicable)
       reduce $\min$ over predecessor states $s$
       write $J_{t+1}^\omega(s')$
 6: **end for**
 7: **return** All $J_t^\omega$

---

## A.3 Instantiation A: Split DP on a Giant Tour in the Vehicle Routing Problem with Stochastic Demand.

**Problem Statement.** The stochastic programming community has extensively studied scenario-based formulations, where uncertainty is modeled by a finite set of realizations. However, scenario-based evaluation quickly becomes computationally prohibitive on CPUs, where even tens of thousands of scenarios can overwhelm multi-threaded implementations. In our work, we adopt this

*scenario-based modeling* framework, in which customer demands are represented by sampled realizations. This approach naturally accommodates correlated demand structures and supports data-driven modeling when historical records are available. This scenario-based approach falls naturally into the two-stage stochastic programming paradigm, whose general form is: $\min_{x \in X} f_1(x) + \mathbb{E}_\xi [f_2(x, \xi)]$, where $x$ denotes the first-stage routing decisions and the corresponding cost $f_1(x)$, $\xi$ is a random vector representing a realization of customer demands, and $f_2(x, \xi)$ denotes the second-stage recourse cost under each scenario $\xi$.

In our two-stage stochastic optimization problem, the first stage determines the visiting sequence of customers, commonly referred to as a *giant tour* in the context of genetic algorithms (Vidal et al., 2012). This representation encodes a solution as a permutation of customers, from which feasible vehicle routes can be recovered via a split operator under fixed vehicle capacity and scenario-dependent customer demands. We assume *full demand revelation prior to the second stage*, enabling the plan to adapt to the realized scenario. Once demands are known, the giant tour is *split into feasible routes* such that the demand on each route does not exceed vehicle capacity. Thus, given a fixed giant tour (i.e., the visiting sequence), the second-stage evaluation is computationally simple: it only requires splitting the tour according to realized demands. **The objective is to determine a first-stage giant tour that minimizes the expected total travel cost across all scenarios.**

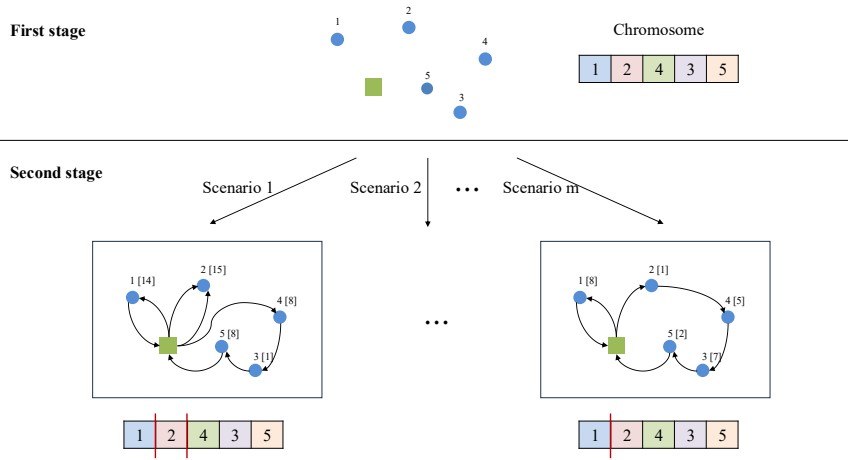

Figure 9: Example of splitting a giant tour into feasible routes under a given demand scenario.

The extensive model is:

$$\min_{x, z, u, l} \quad \frac{1}{|\Omega|} \sum_{\omega \in \Omega} \left( \sum_{k \in \mathcal{K}} \sum_{(i,j) \in \mathcal{A}} (\delta_{ij} + S_j) x_{ijk}^\omega + \beta \sum_{k \in \mathcal{K}} l_k^\omega \right) \tag{5}$$

$$\text{s.t.} \quad \sum_{k \in \mathcal{K}} \sum_{i \in \mathcal{N}} x_{ijk}^\omega = 1 \qquad\qquad \forall j \in \mathcal{N} \setminus \{0\}, \ \forall \omega \tag{6}$$

$$\sum_{k \in \mathcal{K}} \sum_{j \in \mathcal{N}} x_{ijk}^\omega = 1 \qquad\qquad \forall i \in \mathcal{N} \setminus \{0\}, \ \forall \omega \tag{7}$$

$$\sum_{i \in \mathcal{N}} x_{ijk}^\omega = \sum_{i' \in \mathcal{N}} x_{ji'k}^\omega \qquad\qquad \forall j \in \mathcal{N} \setminus \{0\}, \ \forall k \in \mathcal{K}, \ \forall \omega \tag{8}$$

$$\sum_{j \neq 0} x_{0jk}^\omega = \sum_{i \neq 0} x_{i0k}^\omega \leq 1 \qquad\qquad \forall k \in \mathcal{K}, \ \forall \omega \tag{9}$$

$$\sum_{k \in \mathcal{K}} \sum_{j \neq 0} x_{0jk}^\omega \leq K \qquad\qquad \forall \omega \tag{10}$$

$$\sum_{i \in \mathcal{N}} \sum_{j \in \mathcal{N}} x_{ijk}^{\omega} D_j^{\omega} - C \leq l_k^{\omega} \qquad \forall k \in \mathcal{K},\ \forall \omega \tag{11}$$

$$u_{ik}^{\omega} - u_{jk}^{\omega} + 1 \leq n\big(1 - x_{ijk}^{\omega}\big) \qquad \forall i \neq j,\ i,j \geq 1,\ \forall k,\ \forall \omega \tag{12}$$

$$\sum_i z_{ij} + Y_{ij} = 1 \qquad \forall j \geq 1 \tag{13}$$

$$\sum_j z_{ij} + Y_{ij} = 1 \qquad \forall i \geq 1 \tag{14}$$

$$z_{ij} \geq \sum_k x_{ijk}^{\omega} \qquad \forall i > 0, j > 0,\ \forall \omega \tag{15}$$

$$z_{ij} \geq \sum_k x_{i0k}^{\omega} + \sum_k x_{0jk}^{\omega} - 1 - \zeta_{ij} \qquad \forall i > 0, j > 0,\ \forall \omega \tag{16}$$

$$\zeta_{ij} \geq U_j - U_i - 1 \qquad \forall i, j \tag{17}$$

$$\zeta_{ij} \geq U_i - U_j + 1 \qquad \forall i, j \tag{18}$$

$$U_i - U_j + 1 \leq n(1 - z_{ij}) \qquad \forall i \neq j,\ i,j \geq 1 \tag{19}$$

$$\sum_i \sum_j Y_{ij} = 1 \tag{20}$$

$$x_{ijk}^{\omega} \in \{0,1\} \qquad \forall i \neq j,\ \forall k,\ \forall \omega \tag{21}$$

$$z_{ij} \in \{0,1\} \qquad \forall i \neq j \tag{22}$$

$$1 \leq u_{ik}^{\omega} \leq n \qquad \forall i, k, \omega \tag{23}$$

$$l_k^{\omega} \geq 0 \qquad \forall k, \omega \tag{24}$$

Suppose the first-stage giant tour is $(1, 2, 4, 3, 5)$ with vehicle capacity $Q = 17$, as illustrated in Figure 9.

  (i) In Scenario 1, the realized demands are $[14, 15, 8, 1, 8]$. The second-stage routing may then split the tour into three feasible routes: $(1)$, $(2)$, and $(4, 3, 5)$.

  (ii) In Scenario $m$, the realized demands are $[8, 1, 5, 7, 2]$ with three vehicles available. The second-stage routing may then split the tour into two feasible routes: $(1)$ and $(2, 4, 3, 5)$.

It is well understood that robust first-stage policy search benefits from incorporating a sufficiently large number of demand scenarios to capture the breadth of uncertainty; see Shapiro (2003) and Appendix A.5. Theoretical results show that a sufficiently large scenario set is critical for achieving accurate and stable estimates. However, in practice, evaluating even tens of thousands of scenarios can be prohibitively expensive, especially for combinatorial problems such as the CVRPSD, since each scenario requires solving a non-trivial routing evaluation (typically through a dynamic programming algorithm with time complexity $O(nP)$, where $n$ is the number of customers and $P$ is the number of possible transitions) rather than a simple function call.

**Pseudo-code for 2D Kernel.** In this section, we describe the GPU kernel designed to generate data splits for each scenario in Algorithm 3. In Algorithm 2, we launch this kernel with a grid of thread blocks to materialize the output matrix Splits for all scenarios are produced concurrently, which enables simultaneous split generation across all scenarios.

**Illustrative Matrix Form of DP.** To clarify the structure of the transition cost matrix $A^{\omega}$ in the split problem, consider a simple instance with a giant tour $\sigma = [\sigma_1, \sigma_2, \sigma_3]$, and realized demands under scenario $\omega$ given by:

$$(q_{\sigma_1}^{\omega}, q_{\sigma_2}^{\omega}, q_{\sigma_3}^{\omega}) = (2, 3, 4), \quad \text{with vehicle capacity } Q = 5.$$

Let $J^{\omega}(i)$ denote the minimum cost to serve customers $\sigma_1$ through $\sigma_i$. We construct a transition cost matrix $A^{\omega} \in (\mathbb{R} \cup \{+\infty\})^{4 \times 4}$, indexed by prefix states $p, i \in \{0, 1, 2, 3\}$, where the $(p, i)$ entry

---

**Algorithm 2** GPU Splitting Algorithm

---

**Input:** Scenarios $\omega \in \Omega$
**Input:** Global settings $Q, C$ for vehicle capacity and travel cost matrix.
**Output:** Running costs $V$
 1: Initialize $V \in \mathbb{R}^{n \times |\Omega|}$ to all zero matrix
 2: Instantiate multiple 2D kernels in parallel for $V^\omega$.
 3: $V = [V^{\omega_0}, V^{\omega_1}, ..., V^{\omega_{|\Omega|}}]$
 4: **return** $V$

---

**Algorithm 3** 2D Kernel for Splitting

---

**Input:** Scenario Index $\omega$
**Input:** Scenario-specific demand $q^\omega$
**Input:** Global settings $Q, C$ for vehicle capacity and travel cost matrix.
**Output:** Running costs $V^\omega$
 1: Initialize $V^\omega$ to all zero vectors
 2: Initialize $\zeta^\omega$ to an empty queue.
 3: **for** each customer $i$ **do**
 4:     Update potential $V_{(i)}^\omega$
 5:     **if** $i < n$ **then**
 6:         **if** $i$ dominates $\zeta_{\text{back}}^\omega$ **then**
 7:             **while** $\zeta^\omega \neq \emptyset$ AND $\zeta_{\text{back}}^\omega$ dominates $i$ from right **do**
 8:                 Pop $\zeta^\omega$ from back
 9:             **end while**
10:             Push $i$ to $\zeta^\omega$ from back
11:         **end if**
12:         **while** $|\zeta^\omega| > 1$ AND $\zeta_{\text{front}}^\omega$ better than $\zeta_{\text{next\_front}}^\omega$ **do**
13:             Pop $\zeta^\omega$ from front
14:         **end while**
15:     **end if**
16: **end for**
17: **return** $V^\omega$

---

represents the cost of serving customers $\sigma_{p+1}$ to $\sigma_i$ in one route, if the cumulative demand is within capacity. Entries with $p \geq i$ are not applicable and are denoted by x.

Assume the travel cost matrix is:

$$[c_{a,b}] = \begin{bmatrix} 0 & 1 & 2 & 3 \\ 1 & 0 & 1 & 2 \\ 2 & 1 & 0 & 1 \\ 3 & 2 & 1 & 0 \end{bmatrix},$$

where node 0 is the depot and, for simplicity, $\sigma_k$ is identified with node $k$. Each route starts and ends at node 0. Then we compute:

- $A^\omega(0, 1)$: route = $[0, \sigma_1, 0]$, demand = $2 \leq 5$: feasible. Cost = $c_{0,1} + c_{1,0} = 1 + 1 = 2$.
- $A^\omega(0, 2)$: route = $[0, \sigma_1, \sigma_2, 0]$, demand = $5 \leq 5$: feasible. Cost = $c_{0,1} + c_{1,2} + c_{2,0} = 1 + 1 + 2 = 4$.
- $A^\omega(0, 3)$: route = $[0, \sigma_1, \sigma_2, \sigma_3, 0]$, demand = $9 > 5$: infeasible. Cost = $+\infty$.
- $A^\omega(1, 2)$: route = $[0, \sigma_2, 0]$, demand = $3 \leq 5$: feasible. Cost = $c_{0,2} + c_{2,0} = 2 + 2 = 4$.
- $A^\omega(1, 3)$: route = $[0, \sigma_2, \sigma_3, 0]$, demand = $7 > 5$: infeasible. Cost = $+\infty$.
- $A^\omega(2, 3)$: route = $[0, \sigma_3, 0]$, demand = $4 \leq 5$: feasible. Cost = $c_{0,3} + c_{3,0} = 3 + 3 = 6$.

Thus, the masked transition matrix becomes:

$$A^\omega = \begin{bmatrix} \text{x} & 2 & 4 & +\infty \\ \text{x} & \text{x} & 4 & +\infty \\ \text{x} & \text{x} & \text{x} & 6 \\ \text{x} & \text{x} & \text{x} & \text{x} \end{bmatrix}.$$

The forward DP proceeds *column by column* from the initialization $J^\omega(0) = 0$:

$$\boxed{J^\omega(1) = J^\omega(0) + A^\omega(0,1) = 0 + 2 = 2}$$

since only predecessor $p = 0$ is admissible when $i = 1$. Then,

$$\boxed{J^\omega(2) = \min\left\{J^\omega(1) + A^\omega(1,2),\ J^\omega(0) + A^\omega(0,2)\right\} = \min\{2+4,\ 0+4\} = 4}.$$

Finally,

$$\boxed{J^\omega(3) = \min\left\{J^\omega(2) + A^\omega(2,3),\ J^\omega(1) + A^\omega(1,3),\ J^\omega(0) + A^\omega(0,3)\right\} = \min\{4+6,\ 2++\infty,\ 0++\infty\} = 10}.$$

Thus, the running costs after this pass are

$$V^\omega = \left(J^\omega(0), J^\omega(1), J^\omega(2), J^\omega(3)\right) = (0,\ 2,\ 4,\ 10).$$

This illustrates the left-to-right split recursion: each new $J^\omega(i)$ is obtained by minimizing over the previous prefixes,

$$J^\omega(i) = \min_{p<i}\{J^\omega(p) + A^\omega(p,i)\}.$$

The same recursion naturally extends to larger giant tours by advancing $i = 1, 2, \ldots, n$.

### A.4 EVALUATION ON INSTANTIATION B: FORWARD INVENTORY REINSERTION DP IN DYNAMIC STOCHASTIC INVENTORY ROUTING PROBLEMS.

**Problem Statement.** We consider a two-stage stochastic version of the Inventory Routing Problem (IRP) under an Order-Up-To (OU) inventory policy. We define the model on a complete directed graph $\mathcal{G} = (\mathcal{N}, \mathcal{A})$, where $\mathcal{N} = \{0, n+1\} \cup \mathcal{N}'$ includes the supplier node 0, the destination depot $n+1$, and the set of customers $\mathcal{N}'$. The arc set $\mathcal{A}$ represents all possible directed connections between nodes. A set of vehicles $\mathcal{K}$ is available for deliveries, with $|\mathcal{K}| = K$. The planning horizon spans a finite set of days $\mathcal{T} = \{1, \ldots, H\}$ and $\mathcal{T}' = \{2, \ldots, H\}$ for the second stage. To model uncertainty in demand, we consider a finite set of scenarios $\Omega$, where each scenario $\omega \in \Omega$ occurs with probability $p^\omega$. Each vehicle has a capacity limit $Q$, and each customer $i \in \mathcal{N}'$ has an inventory capacity $U_i$. Holding costs $h_i$ are incurred per unit of inventory stored at node $i$, and a stock-out penalty is incurred at a rate of $\rho h_i$ per unit of unmet demand at customer $i$, where $\rho > 1$. The customer demand at customer $i$ on day $t$ under scenario $\omega$ is denoted by $d_i^{t,\omega}$. The cost of traveling from node $i$ to node $j$ is denoted $c_{ij}$ for each arc $(i,j) \in \mathcal{A}$. Finally, $I_i^0$ represents the initial inventory level at node $i$ at the beginning of the planning horizon. To simplify replenishment decisions and reflect common logistics practices, we adopt an OU policy. Under this policy, each customer is either replenished up to its full capacity $U_i$, or not replenished at all on a given day. This is modeled using binary variables $z_i^t$ (or $z_i^{t,\omega}$ in the second stage), which equal 1 if customer $i$ is replenished on day $t$, and 0 otherwise.

The first-stage of the 2SIRP involves determining the vehicle routing and delivery quantities on Day 1, before the actual demand realizations are observed. Given the first-stage decision and realized demand $\boldsymbol{d}^\omega = \{d_i^{t,\omega}\}_{i\in\mathcal{N}, t\in\mathcal{T}'}$ under scenario $\omega$, the second-stage cost function $\tilde{Q}^\omega(x)$ can be calculated as $h_0 I_0^1 + \sum_{k\in\mathcal{K}} \sum_{(i,j)\in\mathcal{A}} c_{ij} y_{ij}^{k,1} + \sum_{\omega\in\Omega} p^\omega \tilde{Q}^\omega(x)$. The second-stage constraints under the OU policy govern the evolution of inventory, replenishment quantities, and routing decisions from Day 2 to Day $H$ under each demand scenario $\omega \in \Omega$. Constraints enforce the OU policy: a customer either receives a shipment that raises its inventory up to the full capacity $U_i$ (i.e., $q_i^{k,t,\omega} = U_i - I_i^{t-1,\omega}$), or receives nothing. Constraints also need to ensure other classical IRP constraints, such as the total quantity delivered by any vehicle $k$ on day $t$ does not exceed its capacity $Q$.

**Pseudo-code for 3D Kernel.** To efficiently adapt the CPU-based DP algorithm for GPUs, we design a two-part framework. Specifically, Algorithm 4 selects a GPU-feasible batch size, solves scenarios in batches via a DP kernel, and aggregates results, reducing the batch size and retrying upon out-of-memory. Algorithm 5 executes a batched backward dynamic program over the horizon—parallel across scenarios and states—comparing "no delivery" versus "deliver to capacity," recording the minimizer and transition, and then backtracking from each scenario's initial inventory to recover daily decisions, quantities, and total cost.

---

**Algorithm 4** GPU-Based DP with Adaptive Batching

---

**Input:** `params`, `client_id`, `all_scenarios_data`, initial batch size $B_0$, device
**Output:** List of scenario results (cost, decision flags, quantities)
 1: $B \leftarrow$ ADJUSTBATCHSIZE($B_0$, device, `params`)  ▷ based on free GPU memory
 2: `results` $\leftarrow []$; `start` $\leftarrow 0$; $N \leftarrow$ #scenarios
 3: **while** `start` $< N$ **do**
 4:     `batch` $\leftarrow$ slice(`all_scenarios_data`, `start:start+B`)
 5:     (`costs, flags, qty`) $\leftarrow$ SOLVEBATCHDP(`batch`, `params`, device)
 6:     Append per-scenario tuples to `results`
 7:     `start` $\leftarrow$ `start` $+B$
 8:     CLEARCACHE()
 9: **end while**
10: **return** `results`

---

**Algorithm 5** SOLVEBATCHDP

---

 1: **function** SOLVEBATCHDP(`batch`, `params`, device)
 2:     Preprocess tensors (time-major, contiguous, to device); set horizon $T$, state grid $\mathcal{S}$
 3:     Initialize DP arrays $C, D, P$; set $C[T] \leftarrow 0$
 4:     **for** $t = T - 1, \dots, 0$ **do**  ▷ (parallel over scenarios/states on GPU)
 5:         Compute no-delivery cost and next state
 6:         Compute deliver-to-max cost (fixed + capacity + holding) and next state
 7:         $C[t] \leftarrow \min(\cdot)$; $D[t] \leftarrow$ argmin decision; $P[t] \leftarrow$ next-state index
 8:     **end for**
 9:     **return** BACKTRACK($C, D, P$, start inventories)  ▷ costs, flags, quantities
10: **end function**

---

**Illustrative Matrix Form of DP.** Consider a single customer with $U_i = 2$, horizon $t = 1, 2$, and a single scenario $\omega$. The initial inventory is $I_i^0 = 1$, daily demand is $d_i^{t,\omega} = 1$, the transportation cost is $F_t(q) = q$, and the end-of-day inventory cost is

$$h_i^t(I) = \begin{cases} 0, & I = 2, \\ 1, & I = 1, \\ 5, & I = 0. \end{cases}$$

The state space each day is $\mathcal{S}_i^t = \{0, 1, 2\}$.

**Day 1.** The forward transition matrix $A_i^{1,\omega}(I, J)$ (rows $I$, columns $J$) is

$$A_i^{1,\omega} = \begin{bmatrix} 5 & 3 & +\infty \\ 5 & 2 & +\infty \\ +\infty & 1 & +\infty \end{bmatrix}, \quad (q \in \{0, U_i - I\},\ J = \max\{0, I + q - 1\}\ ).$$

The initial cost vector encodes $I_i^0 = 1$:

$$J_i^1 = \begin{bmatrix} +\infty \\ 0 \\ +\infty \end{bmatrix}.$$

The column-wise min–plus update gives

$$J_i^2(J) = \min_{I \in \{0,1,2\}} \{J_i^1(I) + A_i^{1,\omega}(I, J)\}, \quad J_i^2 = \begin{bmatrix} \min\{\infty+5,\ 0+5,\ \infty\} \\ \min\{\infty+3,\ 0+2,\ \infty+1\} \\ \min\{\infty,\ \infty,\ \infty\} \end{bmatrix} = \begin{bmatrix} 5 \\ 2 \\ +\infty \end{bmatrix}.$$

**Day 2.** Parameters are the same, so $A_i^{2,\omega} = A_i^{1,\omega}$. Updating once more:

$$J_i^3(J) = \min_I \{J_i^2(I) + A_i^{2,\omega}(I, J)\}, \quad J_i^3 = \begin{bmatrix} \min\{5+5,\ 2+5,\ \infty\} \\ \min\{5+3,\ 2+2,\ \infty+1\} \\ \min\{\infty,\ \infty,\ \infty\} \end{bmatrix} = \begin{bmatrix} 7 \\ 4 \\ +\infty \end{bmatrix}.$$

With a two-day horizon, the terminal cost is $\min_J J_i^3(J) = 4$.

**Policy insight.** - Day 1: from $I = 1$, delivering $q = 1$ (replenish to full) is optimal, leading to $J = 1$ with cost $1 + 1 = 2$. - Day 2: again from $I = 1$, delivering $q = 1$ is optimal, adding another 2. - Total cost = $2 + 2 = 4$, which matches $J_i^3$. If Day 1 skips delivery, $I$ drops to 0 (cost 5), and even if Day 2 delivers 2, the total becomes 8, which is suboptimal.

*Notes.* (i) The third column of $A$ is always $+\infty$ because with $d = 1$ the end-of-day inventory cannot exceed 1. (ii) In general, if evaluating $A(I, J)$ requires enumerating multiple route options $r \in \mathcal{K}$, then $A(I, J) = \min_r A(I, J; r)$, and GPU parallelism naturally extends to three dimensions $(\omega, I{\to}J, r)$ with reductions first over $r$ and then over $I$.

## A.5 MONTE CARLO METHOD PROPOSITION.

*Empirical Risk Minimization* (ERM) method is analogous to the Monte Carlo method for estimating a population mean via sample averages and fits naturally within the ERM framework for stochastic programming. Formally, we distinguish between:

- *True problem*:
$$(P) \quad z^* = \min_{x \in \mathbb{X}} \mathbb{E}[f(x, \tilde{\xi})],$$

- *Sample-average problem* with $m$ scenarios:
$$(P_m) \quad z_m^* = \min_{x \in \mathbb{X}} \frac{1}{m} \sum_{i=1}^m f(x, \tilde{\xi}^i),$$

As the sample size $m$ increases, the optimal solution $x_m^*$ of the sample problem converges to the true optimal solution $x^*$, and the optimal value $z_m^*$ approaches $z^*$. The following result summarizes the fundamental properties of the ERM method under mild regularity conditions (cf. (Shapiro, 2003)).

**Proposition 1.** *Let $\{\tilde{\xi}^1, \ldots, \tilde{\xi}^m\}$ be i.i.d. samples of $\tilde{\xi}$. Denote by*

$$(P) \quad z^* = \min_{x \in \mathbb{X}} \mathbb{E}[f(x, \tilde{\xi})], \qquad (P_m) \quad z_m^* = \min_{x \in \mathbb{X}} \frac{1}{m} \sum_{i=1}^m f(x, \tilde{\xi}^i),$$

*the true and sample average problems, respectively, with optimal solutions $x^*$ and $x_m^*$. Then:*

$$
\begin{array}{lll}
\text{(Bias)} & \mathbb{E}[f(x_m^*, \tilde{\xi})] \leq z^*, & (25) \\[2mm]
\text{(Consistency)} & \mathbb{E}[f(x_m^*, \tilde{\xi})] \xrightarrow{a.s.} z^* \quad \text{as } m \to \infty, & (26) \\[2mm]
\text{(Probabilistic Convergence)} & \lim_{m \to \infty} \Pr\left\{ \mathbb{E}[f(x_m^*, \tilde{\xi})] \leq z^* + \tilde{\epsilon}_m \right\} \geq 1 - \alpha, \quad \tilde{\epsilon}_m \downarrow 0, & (27) \\[2mm]
\text{(Rate of Convergence)} & \sqrt{m}\,(z_m^* - z^*) \xrightarrow{d} \mathcal{N}\!\left(0, \sigma^2(x^*)\right). & (28)
\end{array}
$$

## A.6 GPU-BASED SPLIT VS. BASELINE MILP ON CVRPSD

We use the state-of-the-art MILP solver Gurobi to solve the CVRPSD extensive form under varying numbers of scenarios and customers, and compare its performance with our GPU-based splitting algorithm. The results presented in Table 1 show a clear scalability gap: while GPUSPLIT's runtime grows roughly linearly with the number of scenarios, the CPU-based MILP quickly becomes impractical, hitting time limits and eventually exhausting memory. In addition, for larger scenario sets, GPUSPLIT remains tractable and often achieves better objective values. For example, on the 10-customer dataset with over 1,000 scenarios, neither method proves optimality, but GPUSPLIT consistently finds higher-quality solutions than the CPU-based solver.

## A.7 GPU REINSERTION VS. BASELINE ALGORITHM ON DSIRP

Regarding DSIRP, we provide a comparison with the state-of-the-art method in Coelho et al. (2012), which solves DSIRP under a single scenario. As shown in Table 2, across instances of various sizes,

Table 1: Comparison of GPUSPLIT and the MILP solver on CVRPSD instances with different numbers of scenarios, reporting runtimes, objective values, and failure cases (time limits and OOM).

| | | | #Scenario | | | | |
|---|---|---|---|---|---|---|---|
| | | | 100 | 500 | 1000 | 2000 | 10000 |
| 6 Customer | Time (s.) | GPUSPLIT | 13.76 | 28.92 | 52.64 | 95.42 | 420.11 |
| | | MILP | 26.7 | 587.33 | 1708.51 | **3600** | **OOM** |
| | Obj. Value | GPUSPLIT | 3399.59 | 3392.72 | 3391.86 | 3391.43 | 3391.17 |
| | | MILP | 3399.59 | 3392.72 | 3391.86 | **10541.3** | - |
| 10 Customer | Time (s.) | GPUSPLIT | 485.03 | 3215.46 | **3600** | **3600** | **3600** |
| | | MILP | **3600** | **3600** | **3600** | **3600** | **OOM** |
| | Obj. Value | GPUSPLIT | 3797.37 | 3804.89 | 3800.62 | 3802.95 | 3955.16 |
| | | MILP | 3797.37 | 3804.89 | **3913.01** | **5336.41** | - |

our approach achieves 4.21%-6.37% lower objective values (5.09% on average), while keeping runtimes in the same order of magnitude and even running faster on the largest class. This demonstrates that explicitly handling multiple demand scenarios can yield consistently higher-quality DSIRP solutions without incurring prohibitive additional computation compared to Coelho et al. (2012).

Table 2: Performance comparison between the baseline DSIRP algorithm and our multi-scenario method in terms of objective value, runtime, and percentage improvement.

| Instance | Baseline Obj. | Baseline Time (s) | Ours Obj. | Ours Time (s) | Gain (%) |
|---|---|---|---|---|---|
| small ($n < 50$) | 9131.50 | 67.10 | 8703.20 | 102.30 | 4.68 |
| medium ($50 \leq n \leq 100$) | 30137.80 | 888.30 | 28219.00 | 1029.80 | 6.37 |
| large ($n > 100$) | 60051.40 | 9248.90 | 57523.30 | 7200.00 | 4.21 |
| Average | 33106.87 | 3401.40 | 31481.83 | 2777.40 | 5.09 |

## A.8 SOLUTION QUALITY VERSUS THE NUMBER OF EVALUATED FIRST-STAGE CANDIDATE SOLUTIONS

Regarding the number of candidates evaluated, we have conducted experiments to address this question, and present the numerical results in Table 3. The quality of first-stage decisions over a varied number of candidates is an important consideration. We have presented partial results in Figure 6: as the heuristic runs longer, it discovers better first-stage solutions, which can be interpreted as evaluating a larger and more diverse set of candidates. To make this point more explicit, we present an additional set of experiments on the DSIRP instance, which directly reports the evaluation of different first-stage solutions and their corresponding performance under large scenario sets.

Table 3: Relationship between explored candidates and objective improvement in the DSIRP dataset.

| Time (s.) | GPU Calls | # Candidates | Obj. | Gain |
|---|---|---|---|---|
| 120 | 1,380 | **92** | 77,898,383 | - |
| 240 | 2,715 | **181** | 77,840,004 | 0.07% ↓ |
| 360 | 4,065 | **271** | 77,804,713 | 0.12% ↓ |
| 480 | 5,415 | **361** | 77,741,966 | 0.20% ↓ |
| 600 | 6,750 | **450** | 77,741,966 | 0.20% ↓ |

The candidate first-stage decisions are generated by a genetic algorithm and evaluated via GPU, where each GPU call solves a 500-scenario DP subproblem in parallel. Based on the result in Table 3, on the 15-customer, 500-scenario instance, this results in 92–450 evaluated candidates depending on the time budget, during which the framework consistently finds equal or better solutions as more candidates are considered. The reported results clearly show how solution quality evolves

with the number of evaluated candidates, thereby directly illustrating the behavior and viability of our approach under varying candidate counts.

Table 4: Relationship between explored candidates and objective improvement in the CVRPSD dataset.

| Time (s.) | GPU Calls | # Candidates | Obj. | Gain |
|---|---|---|---|---|
| 120 | 304,740 | **304,740** | 3,893.32 | - |
| 240 | 611,841 | **611,841** | 3,811 | 2.11%↓ |
| 360 | 914,889 | **914,889** | 3,800.62 | 2.38%↓ |
| 480 | 1,216,803 | **1,216,803** | 3,800.62 | 2.38%↓ |
| 600 | 1,524,407 | **1,524,407** | 3,800.62 | 2.38%↓ |

We conducted the same set of experiments for CVRPSD (Table 4) and observe the similar pattern as in DSIRP: as the time budget increases and more first-stage candidates are explored, the framework consistently returns equal or better solutions, with no degradation in solution quality. The experimental results clearly track how the objective improves with the number of evaluated candidates, further confirming the robustness and viability of our candidate-based GPU evaluation scheme.

