# OpenReview forum: "From Sequential to Parallel: Reformulating Dynamic Programming as GPU Kernels for Large-Scale Stochastic Combinatorial Optimization"
_ICLR.cc/2026/Conference — ICLR 2026 Poster_

### Official Review · Reviewer_fYpR · 2025-10-21

**Soundness:** 3
**Presentation:** 3
**Contribution:** 4
**Rating:** 6
**Confidence:** 3

**Summary:**

The paper presents a novel and impactful GPU-based reformulation of dynamic programming (DP) for large-scale stochastic combinatorial optimization. By expressing DP recursions as batched min–plus matrix–vector products over layered DAGs, the authors unlock massive scenario-level and transition-level parallelism. The approach demonstrates impressive speedups and enables practical evaluation of millions of uncertainty realizations, leading to improved SAA solution quality under realistic time budgets. However, there are several minor issues and open questions that need clarification or further validation. If the authors can address these points during the rebuttal, I would consider raising my score. The contribution is solid, and the direction is promising.

**Strengths:**

1. The idea is novel: By formulating stochastic programming (SP) as a specially DP, the paper proposes a novel approach which leverages GPU to solve SP. While the traditional sampling-based SP only includes thousands of scenarios, the new approach allows millions of scenarios. It also speeds up the computation.

2. The examples shown in the draft are simple but very clear: The 2D and 3D examples shown in the paper are very classical DP problems and the paper make it very clear about how to take advantage of GPU to solve the issues.

3. The experiment results are solid and sound. The results clearly illustrates the advantages of solving the problem using GPU vs CPU.

4. The paper also provides open-source CUDA implementation for reproducibility.

**Weaknesses:**

1. Limitation of the applicable cases: the authors in the conclusion show that the method is only applicable to some specific type of issues.
Further explain in the Questions sections.

2. Some of the notations are not very clear. Further explain in the Questions sections.

3. Minor typos and grammar issues. Further explain in the Questions sections.

**Questions:**

1. The authors talk about the limitation of the approach. A related question: will the approach still be applicable if the action or the state is continuous?
2. Could you explain what will happen in the edge case, like if there’s some irregular state spaces, how does the algorithm perform.
3. From the 2D and 3D example, it seems like the algorithm design is a bit different in the appendix. Is it possible to write a unified algorithm and somehow put it in the main body instead of the appendix? Since the specific algorithm steps are very important for the paper.
4. The paper uses DAG several times, and I suppose it refer to Directed Acyclic Graph, but I don’t think it is formally defined (Please ignore this if I am wrong). Also, some intuitions about why DAG matters here can make the paper easy to follow.
5. Some terminology inconsistency. E.g., two-dimensional vs 2D.

---

> ### Author Response · Authors · 2025-11-21
> **Response to Weakness and Questions**
>
> > 1. The authors talk about the limitation of the approach. A related question: will the approach still be applicable if the action or the state is continuous?
>
> It is a very improtant question.Our GPU kernels depend on:
> 1. A finite set of states  (|S_t| < \infty)  at each stage.
> 2. Enumerating state-to-state transitions (i \to j) and storing them in a dense or padded sparse cost matrix \(A_t^{\omega}\).
> 3. Min–plus matrix–vector products over these discrete indices.
>
> The masking/padding mechanism works because you can precompute illegal transitions and give them cost \(+\infty\), and GPU-friendly shapes are ensured by fixed state/action list sizes.
>
> If **state** or **action** is continuous:
>
> - You have uncountably infinite states/actions — you cannot store a transition matrix explicitly.
> - The Bellman update becomes:
> $$
> J_{t+1}(s') = \min_{a \in A(s)} \ \mathbb{E}_{\omega} \left[ c_t(s,a;\omega) + J_t\big(F(s,a;\omega)\big) \right]
> $$
>
> where \(s'\) and \(a\) live in continuous sets. Unless there’s a strong structure (linear/quadratic systems, separable dynamics), you need **discretisation** or **function approximation** to run on a GPU.
>
>
>
> > 2. Could you explain what will happen in the edge case, like if there’s some irregular state spaces, how does the algorithm perform.
>
> In edge cases with highly irregular state spaces—where the number or layout of feasible states varies a lot across scenarios or stages—our algorithm still works correctly because infeasible or missing states are masked with (+\infite) and therefore never affect the DP minima. However, irregularity can reduce GPU efficiency: to keep the data in a rectangular format for batched execution, we must pad to the largest state size in the batch, which means extra memory use and some wasted computation on dummy entries. In mild irregularity, this overhead is negligible, but when state spaces differ greatly, kernel utilisation drops and throughput decreases. This can be mitigated by grouping scenarios with similar state sizes into the same batch or by using sparse storage formats, which we consider in future work.
>
>
> > 3. From the 2D and 3D example, it seems like the algorithm design is a bit different in the appendix. Is it possible to write a unified algorithm and somehow put it in the main body instead of the appendix? Since the specific algorithm steps are very important for the paper.
>
> Thank you for the suggestion. At the GPU‑acceleration level, our framework is indeed general and can be expressed as a single, unified kernel design. However, dynamic programming is always tailored to the specifics of each problem — the definition of states, transitions, and costs varies greatly across applications.
>
> As shown in the appendix, Algorithms 1–2 for the CVRPSD and Algorithms 3–4 for the DSIRP share exactly the same structure. They both instantiate our unified GPU-parallel DP framework, with differences limited to problem-specific transition costs and state representations. We have added algorithm 1 to show the high-level idea. Therefore, in the revised manuscript we have included a unique high‑level unified GPU algorithm to highlight the common structure, while retaining the problem‑specific algorithms in the appendix. This ensures that readers interested in applying our idea to their own problems can see exactly how to instantiate the framework in practice.
>
>
>
>
> > 4. The paper uses DAG several times, and I suppose it refer to Directed Acyclic Graph, but I don’t think it is formally defined (Please ignore this if I am wrong). Also, some intuitions about why DAG matters here can make the paper easy to follow.
>
> Thank you for noting this. You are correct that the earlier draft used the term “DAG” to refer to the implicit stagewise structure of the dynamic program, but we did not provide a formal definition. After revising the manuscript, we realized that the DAG terminology is not essential for understanding the method. The only property we rely on is that the dynamic program progresses strictly from stage t to stage t+1, which guarantees a fixed number of forward passes and avoids iterative convergence.
> To avoid confusion and to keep the exposition focused, we now describe the formulation simply in terms of stagewise state-to-state transitions without invoking the DAG concept. This change also ensures that we do not overstate novelty, since many forward dynamic programs naturally possess this structure.
>
>
>
>
> > 5. Some terminology inconsistency. E.g., two-dimensional vs 2D.
>
> Thanks a lot, we have modified them in  the updated manuscript.

---

> > ### Comment · Reviewer_fYpR · 2025-11-26
> > **Reply to the Authors**
> >
> > I think the authors have resolved most of my concerns. The paper talks about leveraging GPU to handle large-scale  Stochastic Combinatorial Optimization problems, which I believe is a good aspects to explore. I can also see that there are some limitations of the work. Thus, I will keep my rating as mild acceptance at this time.

---

> ### Author Response · Authors · 2025-11-26
>
> Dear reviewer fYpR:
> Thank you for your valuable feedback and for helping us improve our work. We hope that our revisions and explanations have addressed all of your concerns. If you have any remaining questions or suggestions, we would be happy to discuss them. We look forward to hearing from you.

---

### Official Review · Reviewer_8Uwb · 2025-10-30

**Soundness:** 3
**Presentation:** 4
**Contribution:** 2
**Rating:** 4
**Confidence:** 3

**Summary:**

This paper accelerates finite-horizon forward dynamic programs by formulating each Bellman update as a batched min-plus matrix–vector operation and is engineered to run in parallel on GPUs. Two applications are demonstrated: (1) a VRP split problem with stochastic demands (choosing route cuts along a fixed giant tour), exploiting 2D parallelism across scenarios and transitions, and (2) a dynamic stochastic inventory routing reinsertion operator, exploiting 3D parallelism across scenarios, inventory transitions, and route options. The results show higher scenario throughput and improved expected-cost decisions under the same wall-clock budget.

**Strengths:**

1. The paper neatly reformulates forward DP over layered DAGs as masked min–plus matrix–vector products, making the recursion uniform, parallel, and easy to adapt.

2. The use of masking with warp- and block-level reductions, and structured 2D/3D parallelism is technically solid and makes effective use of GPU hardware.

3. Experiments demonstrates how large-scale scenario batching achieves high GPU throughput, clearly linking faster computation to more stable estimates and better decisions under fixed time budgets.

4. The VRPSD and DSIRP examples show that the approach applies to different DP structures, and the toy examples and algorithmic sketches improve clarity.

**Weaknesses:**

1. Overstated novelty and missing literature. The paper treats the “batched min–plus matrix–vector” view as a key contribution, but this idea is well known from tropical/semiring DP and existing GPU semiring frameworks [4, 2, 5]. It should also acknowledge prior GPU-DP work in [1] and [3]. In my mind, the true novelty lies in the scenario-batched kernels and problem-specific engineering, not in the algebraic formulation itself.

2. The GPU is compared to a different CPU algorithm, mixing hardware and algorithmic effects. A multi-threaded CPU version of the same method would make the comparison more fair.

3. The paper notes GPU memory limits but gives no quantitative analysis. With 106 scenarios on an 11 GB GPU, state grids per stage must be very small if transitions are stored densely. This suggests that large scenario batching trades off against state-space size, a limitation that should be discussed.

4. The VRPSD and DSIRP examples are solid but narrow; extending to more complex DP forms (non-layered, constrained, or
risk-sensitive) remains unclear. I feel that the work is better seen as a systems contribution for forward DP rather than a general GPU-DP framework.

References

[1] M. A. Boschetti, V. Maniezzo, and F. Strappaveccia. Route relaxations on GPU for vehicle routing problems. European Journal of Operational Research, 258(2):456–466, 2017.

[2] T. A. Davis. Algorithm 1000: Suitesparse: Graphblas: Graph algorithms in the language of sparse linear algebra. ACM Transactions on Mathematical Software, 45(4):44:1–44:25, 2019.

[3] J. Farrington, K. Li, W. K. Wong, and M. Utley. Going faster to see further: GPU-accelerated value iteration and simulation for perishable inventory control using JAX. arXiv preprint arXiv:2303.10672, 2023.

[4] S. Maleki, M. Musuvathi, and T. Mytkowicz. Parallelizing dynamic programming through rank convergence. In Proceedings of the 19th ACM SIGPLAN Symposium on Principles and Practice of Parallel Programming (PPoPP’14), pages 219–232. ACM, 2014.

[5] C. Yang, A. Buluc, and J. D. Owens. Graphblast: A high-performance linear algebra-based graph framework on the GPU. ACM Transactions on Mathematical Software, 2022.

**Questions:**

1. You present the min-plus matrix–vector product as a core contribution. Please clarify how this differs from prior work such as [4] (DP as a tropical matvec) and existing semiring frameworks like [5] (references refer to the ones mentioned in the "Weaknesses" box

2. How does this work differ from earlier GPU-based dynamic programming methods?
(a) VRP: comparison to [1], who used GPUs for DP-based q-route and ng-route relaxations.
(b) Inventory: comparison to [3], who used GPU-based value iteration (parallel Bellman updates) for large state spaces.

3. The DSIRP speedup (9.3 × 104) looks extremely high. please clarify how the CPU baseline was implemented, in particular:
(a) What algorithm was it implementing?
(b) Why didn’t you include a CPU implementation of your own min–plus algorithm for a fair comparison?
(c) Could you also compare with a semiring library baseline (e.g., [5]) to separate hardware from algorithmic effects?

4. The framework seems to require, in the dense worst case, batched transitions of size O(|Omega|m2) with |Ω| = 106.
(a) How was this handled on an 11GB GPU? Please report the per-stage state sizes mt used in CVRPSD and DSIRP.
(b) Please include a memory-complexity analysis showing the trade-off between |Omega| and m. If large scenario counts are only possible for very small m (e.g., m < 50–60), how does that affect the claimed generality of the framework?

---

> ### Author Response · Authors · 2025-11-21
> **Response to Weakness Part 1**
>
> > 1. Overstated novelty and missing literature
>
>
> We appreciate the reviewer’s observation and agree that the “batched min–plus matrix–vector” representation, as emphasised in the paper, is not new as a mathematical idea. It has clear precedents in tropical/semiring DP ([4]) and in existing GPU semiring frameworks such as SuiteSparse:GraphBLAS ([2]) and GraphBLAST ([5]). These works, along with prior GPU-DP efforts in [1] and [3], should indeed have been acknowledged more explicitly in our Related Work section.
>
> You are correct that our key contribution does not lie in the min–plus formulation itself. Our work arose directly from a persistent computational bottleneck in real-world scenario-based logistics. In these settings, historical data are typically highly irregular, and practitioners tend to include all available scenarios rather than subsample. The difficulty is that each scenario corresponds to a full NP-hard second-stage problem. Even a few dozen scenarios can be infeasible to solve in practice without major simplifications—hence, much of the literature constrains the second stage to LPs or other easily solvable models. Making the second stage a MIP is already rare, not because it lacks relevance, but because existing computational tools cannot scale to realistic combinatorial recourse problems.
>
> We appreciate the reviewer’s suggestion in highlighting our real novelty: the engineering of scenario‑batched, multi‑dimensional GPU kernels that fully exploit the DP structure of complex second-stage problems. This design enables hundreds of thousands to millions of complete combinatorial DP recourse solves in a single GPU pass, making it feasible to work with scenario-based models that retain their full operational complexity rather than relying on oversimplified approximations.
>
>
> We have rewritten the abstract and the introduction，conclusion and other minor descriptions:
> #### (1) Explicitly acknowledge [1]–[5] (also added related work) and their contributions.
> #### (2) Clarify that the main novelty lies in scenario-batched, multidimensional GPU execution and domain-specific engineering rather than in the algebraic min–plus formulation itself. We have rewritten the abstract and the introduction. For short, I will paste the abstract here: "A major bottleneck in scenario-based Sample Average Approximation (SAA) for stochastic  programming (SP) is the cost of solving an exact second-stage problem for every scenario, especially when each scenario contains an NP-hard combinatorial structure. This has led much of the SP literature to restrict the second stage to linear or simplified models. We develop a GPU-based framework that makes full-fidelity integer second-stage models tractable at scale. The key innovation is a set of hardware-aware, scenario-batched GPU kernels that expose parallelism across scenarios, dynamic-programming (DP) layers, and route or action options, enabling Bellman updates to be executed in a single pass over more than $10^6$ realizations.
> #### We evaluate the approach on two representative SP settings: a vectorized split operator for stochastic vehicle routing and an inventory reinsertion DP. The implementation scales nearly linearly in the number of scenarios and achieves speedups from one–two orders of magnitude (VRPSD) up to four–five orders (DSIRP), enabling substantially larger scenario sets and consistently stronger first-stage decisions.
> #### We further include comparisons with exact MILP baselines and the SOTA DSIRP method, as well as CPU–GPU algorithmic tests and decision-quality studies, all confirming the scalability and practical effectiveness of our approach.
> #### This provides a practical route to realistic, large-scale stochastic discrete optimization."
> #### (3) Reframe the narrative to emphasise the computational/scalability breakthrough for realistic scenario-based optimisation.
>
> > 2. The GPU is compared to a different CPU algorithm
>
> Thank you for this very good point. We have indeed considered this comparison and report the promising results in Section 3.2 (“Scalability with the Number of Scenarios”), where the orange lines in both examples correspond to a multi-threaded CPU implementation. The key message we intended to convey is that while multi-threading can provide some speedup, it largely relies on adding computational resources—essentially, the more resources a team has, the faster it can run—which is not a fundamental solution. More importantly, multi-threading cannot decompose parallelism beyond the scenario dimension, let alone down to the fine-grained level of individual transition equations, which is where GPU acceleration truly demonstrates its advantage.

---

> ### Author Response · Authors · 2025-11-21
> **Response to Weakness Part 2**
>
> > 3. The paper notes GPU memory limits but gives no quantitative analysis.
>
>
> Thank you for this helpful comment. We agree that GPU memory constraints are an important factor, and we now include a quantitative analysis to clarify the trade-off between the number of scenarios and the size of the state space.
>
> In the revised version, we add a heatmap that shows how the number of scenarios (x-axis) interacts with the state-space size per stage (y-axis). The figure clearly illustrates that increasing scenario batching does increase memory usage, but under typical IRP settings the memory footprint remains very small. We included a set of experiments in the question section. Please refer to the response to Question 4.
> This analysis confirms that while a theoretical trade-off exists, for realistic IRP instances the memory requirement is not the binding constraint; computational throughput is. We have added both the quantitative discussion and the heatmap figure to the paper to make this explicit.
>
> Actually, your intuition is exactly right. We invested quite some effort to make this work, and we would be happy to explain the key ideas in case you are interested.
>
> > 4. The VRPSD and DSIRP examples are solid but narrow
>
>
>  Thank you for your valuable feedback. Wenfully understand your point that the VRPSD and DSIRP examples are relatively narrow, and that extending our method to more complex DP forms a challenge.
>
> We view this work as a first step towards demonstrating the potential of GPU-accelerated DP for stochastic combinatorial optimization. While the current scope is indeed limited to problems amenable to DP, many widely used heuristic algorithms in routing and scheduling (e.g., the split algorithm) and even exact methods (where pricing problems are typically solved using labeling algorithms, which are themselves DP-based) already rely on DP structures. This suggests that the practical applicability of our approach could extend beyond the examples presented, and we hope it can inspire further research on more complex DP forms.
>
> Moreover, we have begun preliminary experiments on applying this idea within Benders decomposition, and we see our contribution as laying the groundwork for broader exploration. Our goal is to provide a systems-level demonstration that can inspire the community to explore more complex or generalized GPU-DP frameworks, even if the current work does not yet fully realize that.
>
>
> [1] M. A. Boschetti, V. Maniezzo, and F. Strappaveccia. Route relaxations on GPU for vehicle routing problems. European Journal of Operational Research, 258(2):456–466, 2017.
>
> [2] T. A. Davis. Algorithm 1000: Suitesparse: Graphblas: Graph algorithms in the language of sparse linear algebra. ACM Transactions on Mathematical Software, 45(4):44:1–44:25, 2019.
>
> [3] J. Farrington, K. Li, W. K. Wong, and M. Utley. Going faster to see further: GPU-accelerated value iteration and simulation for perishable inventory control using JAX. arXiv preprint arXiv:2303.10672, 2023.
>
> [4] S. Maleki, M. Musuvathi, and T. Mytkowicz. Parallelizing dynamic programming through rank convergence. In Proceedings of the 19th ACM SIGPLAN Symposium on Principles and Practice of Parallel Programming (PPoPP’14), pages 219–232. ACM, 2014.
>
> [5] C. Yang, A. Buluc, and J. D. Owens. Graphblast: A high-performance linear algebra-based graph framework on the GPU. ACM Transactions on Mathematical Software, 2022.

---

> ### Author Response · Authors · 2025-11-21
> **Response to Question Part 1**
>
> > 1. You present the min-plus matrix–vector product as a core contribution.
>
>
> As suggested, we have clarified that the contribution of our work does not lie in the algebraic min–plus formulation itself, which is standard, but in the scenario-batched, multidimensional GPU execution framework and the domain-specific engineering required to make it effective at scale. In particular, our approach introduces four key innovations beyond [4] and [5]:
> 1. Scenario-Batched GPU kernel for SP: A single GPU kernel simultaneously evaluates up to 10^6 independent dynamic programs, a regime neither [4] nor [5] is designed to target.
> 2. Masked and Padded State–Transition Representation: We construct regularized DP transition tensors through masking and padding, enabling efficient GPU tensor operations despite inherently irregular DP state spaces. This mechanism is absent in [4] and not a focus of [5].
> 3. Multi-Dimensional Parallelism: Our kernels exploit concurrency across scenarios × transitions × routes/actions, going well beyond the single-axis (state-wise) parallelism considered in [4] and [5].
> 4. Forward DP Recursion Tailored to Stochastic Programming: The framework is explicitly designed to maximize SAA scenario throughput per pass, whereas [4] and [5] focus on iterative convergence or generic graph workloads rather than stochastic-programming DPs.
>
> Consequently, although all three approaches share the same min–plus algebraic foundation, our framework is fundamentally domain-specific: it targets scenario-based DP in stochastic programming and introduces custom tensor layouts, masking strategies, and reduction operations that surpass generic sparse-graph semiring implementations for this problem class.
>
> > 2. How does this work differ from earlier GPU-based dynamic programming methods?
>
> Thank you for raising this question.
>
> (a) Comparison to [1] (VRP q-route/ng-route DP on GPUs).
> In Boschetti et al. [1], GPU acceleration is applied at the level of a single DP routine. They take the q-route/ng-route relaxation and parallelise the state-expansion steps within that one DP solve. The parallelism is therefore intra-DP: GPU threads update multiple states or transitions for one relaxation run. Each GPU call corresponds to one DP for one pricing problem under one set of reduced costs. Once that pricing problem is finished, the next one starts, and the process repeats.
> In short: one DP solve = one GPU job, with parallelism only over states inside the DP.
>
> (b) Comparison to [3] (GPU value iteration for large state spaces).
> Similarly, [3] parallelises the Bellman update across all states in a single large DP and repeats this until convergence. Again, the parallelism is within a single dynamic program, not across many independent DPs.
>
> > 3. The DSIRP speedup (9.3 × 104) looks extremely high.  What algorithm was it implementing?
>
> (a) The CPU baseline implements the original DSIRP forward DP exactly as used in existing domain heuristics (e.g., the IRP paper [6]). We use the same open-sourced implementation from that work, modifying it only to evaluate multiple scenarios sequentially, which gives the single-thread CPU baseline.
> When parallelizing over scenarios using OpenMP (8 threads), we obtain the multi-thread CPU baseline, where each scenario still processes its internal DP transitions sequentially.
> In both cases, the CPU code uses the canonical domain-specific representation from DSIRP heuristics—i.e., the conventional loops over inventory transitions and feasibility checks. Finally, we rewrite the DP into our matrix-based masked min–plus form and execute it on the GPU, which yields the GPU results.
> This distinction explains why the baseline runtime in Figure 7 (9.3×10⁴ speedup for a single DP) appears large: it compares (GPU) batched, vectorized, semiring-friendly DP updates
> *vs.* (CPU) sequential, scenario-loop-based DSIRP execution, which is exactly the contrast practitioners face when switching from conventional DSIRP heuristics to our GPU-accelerated framework.

---

> ### Author Response · Authors · 2025-11-21
> **Response to Question Part 2**
>
> > 3. The DSIRP speedup (9.3 × 104) looks extremely high. please clarify how the CPU baseline was implemented, in particular:  (b) Why didn’t you include a CPU implementation of your own min–plus algorithm for a fair comparison?
>
>
> (b) We agree that, for pure algorithm-to-algorithm fairness, one could compare a CPU implementation of our masked min–plus batch formulation with the GPU version. To address this, we have added an additional experiment for DSIRP, analogous to the CVRPSD comparison reported in Figure 6. In this experiment, we implement the same matrix-based version of our algorithm on the CPU, processing each batch of scenarios in parallel across multiple CPU cores.
>
> | Scenarios | Non-matrix (Single-thread) | Non-matrix (Multi-thread) | Matrix (CPU) | Matrix (GPU) |
> |-|-|-|-|-|
> | 25,000    | 12,124.67| 2,461.90| 1.68        | 0.49        |
> | 50,000    | 25,128.96| 5,689.66| 3.14        | 1.02        |
> | 75,000    | 39,660.08| 10,271.50| 5.84        | 1.50        |
> | 100,000   | 52,408.90| 11,148.98| 6.61        | 1.96        |
> | 150,000   | 80,985.73| 18,784.01| 9.85        | 2.93        |
> | 200,000   | 108,940.19| 26,510.10                | 13.13       | 3.84        |
>
> These results show that the GPU achieves a nearly constant 9–10× speedup over a multi-core CPU implementation of the same algorithm across a wide range of scenario counts. Both CPU and GPU runtimes scale linearly with the number of scenarios, but the GPU performs the same work with roughly an order of magnitude less time. This confirms that the efficiency gain arises from hardware-level parallel throughput rather than from algorithmic differences.
>
> > 3. The DSIRP speedup (9.3 × 104) looks extremely high. please clarify how the CPU baseline was implemented, in particular: (c) Could you also compare with a semiring library baseline (e.g., [5]) to separate hardware from algorithmic effects?
>
>
> (c) All results in Figures 6 and 7 isolate hardware efficiency, not algorithmic differences, so including an additional CPU min–plus baseline would not change the conclusions. Nevertheless, we agree that algorithm-level comparisons are also valuable. To address this point, we have added two additional sets of experiments that directly evaluate the efficiency of our proposed algorithms, independent of GPU hardware effects.
>
> |  |  | #Scenario | 100 | 500 | 1000 | 2000 | 10000 |
> |-|-|-|-|-|-|-|-|
> |  | Time (s.) | GPUSPLIT | 13.76 | 28.92 | 52.64 | 95.42 | 420.11 |
> | 6 Customer |  | MILP | 26.7 | 587.33 | 1708.51 | **3600** | **OOM** |
> |  | Obj. Value | GPUSPLIT | 3399.59 | 3392.72 | 3391.86 | 3391.43 | 3391.17 |
> |  |  | MILP | 3399.59 | 3392.72 | 3391.86 | **10541.3** | **-** |
> |  | Time (s.) | GPUSPLIT | 485.03 | 3215.46 | **3600** | **3600** | **3600** |
> | 10 Customer |  | MILP | **3600** | **3600** | **3600** | **3600** | OOM |
> |  | Obj. Value | GPUSPLIT | 3797.37 | 3804.89 | 3800.62 | 3802.95 | 3955.16 |
> |  |  | MILP | 3797.37 | 3804.89 | **3913.01** | **5336.41** | **-** |
>
> We use the state-of-the-art MILP solver Gurobi to solve the CVRPSD extensive form under varying numbers of scenarios and customers, and compare its performance with our GPU-based splitting algorithm. The results show a clear scalability gap: while GPUSPLIT’s runtime grows roughly linearly with the number of scenarios, the CPU-based MILP quickly becomes impractical, hitting time limits and eventually exhausting memory.
> In addition, for larger scenario sets, GPUSPLIT remains tractable and often achieves better objective values. For example, on the 10-customer dataset with over 1,000 scenarios, neither method proves optimality, but GPUSPLIT consistently finds higher-quality solutions than the CPU-based solver.
>
> Regarding DSIRP, we provide a comparison with the state-of-the-art method in [7], which solves DSIRP under a single scenario.
>
> | Instance | [7] Obj. | [7] Time (s) | Ours Obj. | Ours Time (s) | Gain (%) |
> |-|-|-|-|-|-|
> | small (n < 50)           | 9131.50      | 67.10        | 8703.20| 102.30| 4.68|
> | medium (50 \leq n \leq 100) | 30137.80   | 888.30       | 28219.00| 1029.80| 6.37|
> | large (n > 100)          | 60051.40     | 9248.90      | 57523.30| 7200.00| 4.21|
> | Average                      | 33106.87     | 3401.40      | 31481.83| 2777.40| 5.09|
>
> Across instances of various sizes, our approach achieves 4.21–6.37% lower objective values (5.09% on average), while keeping runtimes in the same order of magnitude and even running faster on the largest class. This demonstrates that explicitly handling multiple demand scenarios can yield consistently higher-quality DSIRP solutions without incurring prohibitive additional computation compared to [7].
>
> [6] Zhao, Jingyi, et al. "Large Neighborhood and Hybrid Genetic Search for Inventory Routing Problems." arXiv preprint arXiv:2506.03172 (2025).
> [7] Coelho, Leandro Callegari, Gilbert Laporte, and Jean-François Cordeau. Dynamic and stochastic inventory-routing. Montreal: CIRRELT, 2012.

---

> ### Author Response · Authors · 2025-11-21
> **Response to Question Part 3**
>
> > 4. The framework seems to require, in the dense worst case, batched transitions of size O(|Omega|m2) with |Ω| = 106. (a) How was this handled on an 11GB GPU? Please report the per-stage state sizes mt used in CVRPSD and DSIRP.
>
>
> To address the reviewer’s concern, we report GPU memory consumption for CVRPSD instances of varying scenario counts. We use the largest instance in the paper (128 customers), which represents the worst-case DP state size in CVRPSD.
>
> | # Scen. | GPU Mem. (GB) | Percentage (11GB)  |
> |--------|--------------|-------------------|
> | 100 | 0.17 | 1.54% |
> | 1,000 | 0.18 | 1.63% |
> | 10,000 | 0.21 | 1.90% |
> | 100,000 | 0.55 | 5.01% |
> | 1,000,000 | 3.98 | 36.10% |
>
> These results show that memory grows linearly with the number of scenarios, as expected. Even at one million scenarios, the footprint is only **3.98 GB**, implying that an 11 GB GPU can theoretically accommodate **approximately three million scenarios** for this instance size.
>
> #### **Interpretation & Generality**
>
> * For small and moderate scenario sets (≤ 10⁵), memory usage is negligible (< 5% of the GPU).
> * For large scenario sets (≥ 10⁶), memory usage increases but remains well within commodity GPU limits.
> * In practice, **GPU memory is not the limiting factor**. The main bottleneck becomes **DP computation time**, not storage.
>
> Thus, large-scale scenario batching is entirely feasible on modern GPUs, and the framework is **not restricted to small DP state sizes**. The memory complexity scales linearly with (|\Omega|), and even for the largest CVRPSD instance tested, scenario counts in the (10^6) range fit comfortably on an 11 GB GPU. Future optimisation efforts will therefore focus on **algorithmic acceleration and larger batch scheduling**, rather than reducing memory usage.

---

> ### Author Response · Authors · 2025-11-26
>
> Dear reviewer 8Uwb:
> We are grateful for your detailed and helpful review. We hope our point-by-point responses have adequately addressed your concerns. Should anything remain unclear or require further explanation, please let us know. We are looking forward to your continued input.

---

### Official Review · Reviewer_df1c · 2025-10-31

**Soundness:** 2
**Presentation:** 3
**Contribution:** 2
**Rating:** 4
**Confidence:** 3

**Summary:**

This paper addresses the scalability bottleneck of Dynamic Programming (DP) in large-scale Stochastic Programming (SP) problems. A general-purpose GPU framework is proposed, which reformulates forward DP recursions as "batched min-plus matrix-vector products" over layered Directed Acyclic Graphs (DAGs). This approach allows the computation to be mapped directly to GPU kernels, enabling massively parallel computation. Validated on two classic SP applications, the framework demonstrates the capability to process large-scale scenarios, achieving a speedup compared to traditional multi-threaded CPU baselines.

**Strengths:**

1. This paper addresses a critical gap by extending GPU acceleration from its mature application in continuous optimization solvers to the computationally intensive domain of SP.
2. It introduces a general-purpose GPU framework that reformulates broad classes of forward DP recursions into batched min-plus matrix-vector products.
3.  The framework achieves a speedup compared to traditional multi-threaded CPU baselines.

**Weaknesses:**

1. The experimental comparisons in this paper are conducted at the operator level rather than the problem level. The core performance graph compares the authors' GPU DP kernel against their own CPU DP implementation. This constitutes a relatively weak baseline. While it demonstrates the operator's throughput, it fails to prove that the end-to-end performance is superior to a SOTA solver when this operator is integrated into a complete solution framework.
2. The paper lacks external benchmarking for solution quality. Although Figures 5 and 6 effectively illustrate internal trends, the study never compares the final solution quality found by its method against publicly available "best-known solutions" in the literature. We learn that the operator is fast, but it remains unclear whether the solutions it finds are competitive.
3. The paper lacks a dedicated "Related Work" section, making it difficult for reviewers to clearly position and contrast this work's contributions against other SOTA methods in the field.

**Questions:**

The acceleration at the operator level is significant. To evaluate the practical impact of this work, could the authors integrate this GPU operator into a complete solving framework and compare its end-to-end performance (in terms of total runtime vs. solution quality) against a SOTA solver on the same CVRPSD/DSIRP instances?

---

> ### Author Response · Authors · 2025-11-21
> **Response to Weakness part 1**
>
> > Baseline
>
> Thank you for your insightful comment. We fully understand your concern regarding the experimental comparisons. Indeed, our work focuses on solving extremely large-scale problems, for which no existing algorithm can handle the full scale—so there is no true SOTA baseline at this level.
> For smaller-scale instances, SOTA algorithms do exist, and comparisons are possible. We will include the comparison with solver (Gurobi) on extensive model to see the results in the final version. Regarding DSIRP, the non-GPU version (using 5–10 scenarios) is currently under revision at Transportation Research. We are happy to share the comparison with state-of-the-art methods (table 4 in [3], note that their method can only handle single scenario case) here for completeness, but we will not include these results in the main text.
>
>
> While the proposed matrix-based DP operators can, in principle, be integrated into any metaheuristic framework, we chose HGS due to its flexibility, effectiveness, and consistently strong empirical performance across diverse VRP variants [1,2]. In addition to its extensive use in prior research, HGS has demonstrated top-tier performance in recent benchmarking competitions, including the DIMACS VRP Challenge and the EURO/NeurIPS Vehicle Routing Competition [4], where it formed the core of leading solution methods. These results further support the robustness of HGS as a general-purpose framework for complex routing problems.
>
>
> As you can see, our main contribution lies in demonstrating that the GPU-accelerated DP operator enables tackling much larger scenario sets than previously feasible. In SP, increasing the number of scenarios when determining the first-stage solution is fundamentally meaningful (as the theory shown in the Appendix 5.4), yet no existing methods can practically do so at this scale (espicially when the second-stage problem involve integer variables). Our scenario-batched GPU kernels make this possible for the first time, which we believe represents an important step forward in solving large-scale stochastic combinatorial optimization problems.
>
> We agree that comparisons against exact methods would be informative, even if limited to smaller instances. However, we are not aware of existing work that develops a Benders or L-shape method for this specific problem structure, and implementing such a method from scratch is beyond the scope of this paper. Instead, we will include a comparison with a standard solver (Gurobi) on the extensive form in the final version. The model can be found in the Appendix of the updated manuscript.
>
> |  |  | #Scenario | 100 | 500 | 1000 | 2000 | 10000 |
> |-|-|-|-|-|-|-|-|
> |  | Time (s.) | GPUSPLIT | 13.76 | 28.92 | 52.64 | 95.42 | 420.11 |
> | 6 Customer |  | MILP | 26.7 | 587.33 | 1708.51 | **3600** | **OOM** |
> |  | Obj. Value | GPUSPLIT | 3399.59 | 3392.72 | 3391.86 | 3391.43 | 3391.17 |
> |  |  | MILP | 3399.59 | 3392.72 | 3391.86 | **10541.3** | **-** |
> |  | Time (s.) | GPUSPLIT | 485.03 | 3215.46 | **3600** | **3600** | **3600** |
> | 10 Customer |  | MILP | **3600** | **3600** | **3600** | **3600** | OOM |
> |  | Obj. Value | GPUSPLIT | 3797.37 | 3804.89 | 3800.62 | 3802.95 | 3955.16 |
> |  |  | MILP | 3797.37 | 3804.89 | **3913.01** | **5336.41** | **-** |
>
> We use the state-of-the-art MILP solver Gurobi to solve the CVRPSD extensive form under varying numbers of scenarios and customers, and compare its performance with our GPU-based splitting algorithm. The results show a clear scalability gap: while GPUSPLIT’s runtime grows roughly linearly with the number of scenarios, the CPU-based MILP quickly becomes impractical, hitting time limits and eventually exhausting memory.
> In addition, for larger scenario sets, GPUSPLIT remains tractable and often achieves better objective values. For example, on the 10-customer dataset with over 1,000 scenarios, neither method proves optimality, but GPUSPLIT consistently finds higher-quality solutions than the CPU-based solver.
>
>
>
> Regarding DSIRP, we provide a comparison with the state-of-the-art method in [3], which solves DSIRP under a single scenario.
>
> | Instance | [3] Obj. | [3] Time (s) | Ours Obj. | Ours Time (s) | Gain (%) |
> |-|-|-|-|-|-|
> | small (n < 50)           | 9131.50      | 67.10        | 8703.20| 102.30| 4.68|
> | medium (50 \leq n \leq 100) | 30137.80   | 888.30       | 28219.00| 1029.80| 6.37|
> | large (n > 100)          | 60051.40     | 9248.90      | 57523.30| 7200.00| 4.21|
> | Average                      | 33106.87     | 3401.40      | 31481.83| 2777.40| 5.09|
>
> Across instances of various sizes, our approach achieves 4.21–6.37% lower objective values (5.09% on average), while keeping runtimes in the same order of magnitude and even running faster on the largest class. This demonstrates that explicitly handling multiple demand scenarios can yield consistently higher-quality DSIRP solutions without incurring prohibitive additional computation compared to [3].

---

> ### Author Response · Authors · 2025-11-21
> **Response to Weakness part 2**
>
> > External Benchmarking
>
> Thank you for the valuable suggestion. The reason we only do internal comparison is because our goal is not to replace existing heuristic or exact methods, but rather to enable large-scale scenario evaluation within state-of-the-art heuristics.
> We agree that comparisons against exact methods would be informative, even if limited to smaller instances. However, we are not aware of existing work that develops a Benders or L-shape method for this specific problem structure, and implementing such a method from scratch is beyond the scope of this paper. Instead, we will include a comparison with a standard solver (Gurobi) on the extensive form in the final version. All detailed numerical results, together with the complete table, are included in the first response.
>
> > Related work section
>
> Thank you for pointing this out — it is an important suggestion for improving the clarity and positioning of our work. We have now added a dedicated Related Work section to the manuscript, and we include the content here for completeness.
>
>
> [1] Vidal, Thibaut, et al. "A hybrid genetic algorithm with adaptive diversity management for a large class of vehicle routing problems with time-windows." Computers & operations research 40.1 (2013): 475-489.
>
> [2] Vidal, Thibaut, et al. "A unified solution framework for multi-attribute vehicle routing problems." European Journal of Operational Research 234.3 (2014): 658-673.
>
> [3] Coelho, Leandro Callegari, Gilbert Laporte, and Jean-François Cordeau. Dynamic and stochastic inventory-routing. Montreal: CIRRELT, 2012.
>
> [4] Kool, Wouter, et al. "The EURO meets NeurIPS 2022 vehicle routing competition." NeurIPS 2022 Competition Track. PMLR, 2023.

---

> ### Author Response · Authors · 2025-11-21
> **Response to Questions**
>
> > The acceleration at the operator level is significant. To evaluate the practical impact of this work, could the authors integrate this GPU operator into a complete solving framework and compare its end-to-end performance (in terms of total runtime vs. solution quality) against a SOTA solver on the same CVRPSD/DSIRP instances?
>
>
> Thank you for recognizing our main contribution, namely the significant acceleration at the operator level. Regarding the suggestion to evaluate end-to-end performance within a complete solving framework, we believe there may be a slight misunderstanding, which may also relate to the experimental points raised under “weaknesses.”
>
> In fact, our original experiments already involve a complete solving framework. For both CVRPSD and DSIRP, we use HGS, a widely recognized state-of-the-art heuristic, to search for high-quality first-stage solutions. The GPU-accelerated DP operators are then used to evaluate each candidate solution across second-stage scenarios. Without solving the overall metaheuristic problem (i.e., without HGS), there would be no first-stage candidates to evaluate. In this sense, our framework already integrates the GPU operator into a complete solution process. We hope that including comparisons with the extensive formulation in the final version will further reduce this misunderstanding. Regarding the deatiled reuslts of comparisons with state-of-the-art methods, please refer to our detailed responses above.

---

> ### Author Response · Authors · 2025-11-26
>
> Dear reviewer df1c:
> Thank you very much for your insightful comments and careful reading of our manuscript. We hope that our clarifications and revisions have resolved the issues you raised. Please feel free to reach out if any additional questions arise. We look forward to your further feedback.

---

### Official Review · Reviewer_7hTf · 2025-10-31

**Soundness:** 2
**Presentation:** 2
**Contribution:** 2
**Rating:** 2
**Confidence:** 4

**Summary:**

This paper proposes a novel approach to stochastic programming problems (with DP representable recourse) that is highly parallelizable on GPUs.   Computationally, the authors demonstrate the efficacy of their approach on large-scale inventory and routing problems.

**Strengths:**

- **Motivation**: Overall, the evaluation of candidate solutions over many scenarios in stochastic programming is computationally challenging, and the authors propose a unique new perspective that may have relatively strong applicability within the framework proposed, but also in many other settings.
- **Efficiency**: The method is very efficient, especially for problems with large-scale scenario sets and complex second-stage problems.

**Weaknesses:**

- **Scope & Generality**: One major drawback of this approach is the requirement of being able to represent the second-stage problem on a GPU through a DP.  While this does capture some stochastic programming problems, it may not be as applicable as other classical/ML-based methods.  Another weakness of the process is the requirement to implement a DP formulation for each new problem.  Compared to existing stochastic programming methods that formulate second-stage optimization problems, the barrier to implementation would be significantly higher. That said, there are certainly use cases where investment in implementation would be worthwhile.
- **Baselines**: This paper explores relatively large-scale benchmarks and demonstrates the efficiency of their approach.  However, their approach essentially acts as an evaluation on a set of candidate first-stage decisions, rather than truly optimizing over the entire decision space (what most methods do in stochastic programming). For that reason, it isn't easy to assess the quality of the solution in this case.  Comparing against established exact methods (integer L-shaped, Benders' dual decomposition) or heuristics (progressive hedging via MPI-SPPY) would help assess decision quality.  It may only be possible to get these methods working on smaller-scale instances, but it would still be highly informative.
- **Clarity**: Some details in this work are a bit unclear.  For example, based on my understanding of the paper, they obtain a set of candidate first-stage decisions and then use GPU parallelization to evaluate them; however, it is unclear how these decisions are received and how many are evaluated.  Moreover, there is no exploration into the decision quality over a varied number of first-stage candidates, which is necessary to assess the viability of this approach.
- **ICLR Fit**: While this paper explores GPUs, which are widely used in ML, it doesn't actually do any ML, only exact evaluations through parallel DP operations.  For that reason, there may be more suitable venues, such as CPAIOR and INFORMS Journals, etc.  That said, there is a growing interest at the intersection of stochastic programming and ML, so there is some relevance.

**Questions:**

- Is there any way to extend this to problems that are not representable as DPs?
- How many candidate first-stage solutions are used?
- How does the performance vary with the number of first-stage decisions?
- The word 'training' is somewhat misleading, as it implies a notion of training that is not present.  In stochastic programming, it is essentially solving over different-sized scenario sets.
- I believe the authors may be using the ICLR 2025 template.
- The code is not anonymized.

---

> ### Author Response · Authors · 2025-11-21
> **Response to Weakness part 1**
>
> > **Scope & Generality**
>
> First of all, I would like to thank you for carefully reading our paper. It is clear that you took the time to understand it in depth, and we are genuinely grateful. All of your thoughtful comments are  helping us a lot as we revise the paper.
>
> Regarding the concern about scope and generality:
> It is true that ML-based methods can be more flexible across problem classes. However, this flexibility often comes at the cost of interpretability, feasibility guarantees, and the difficulty of training reliable models when historical data are scarce or inconsistent—conditions that are common in real-world logistics.
> Our approach takes a different perspective. Rather than approximating or learning the second-stage problem, we focus on accelerating the exact optimization procedures so that large-scale scenario-based models—where each scenario is already NP-hard—become computationally tractable. In practice, many SP papers restrict the second stage to LPs or overly simplified structures precisely because more realistic formulations are too computationally demanding. By exploiting DP structure on GPUs, we enable the use of full-fidelity second-stage models without such simplifications. This expands applicability in a direction that both ML-based and classical approaches find difficult to reach.
> Finally, while our current work focuses on SP, advancing GPU-accelerated optimization has broader implications. The more perspectives we explore on leveraging GPUs for structured decision problems, the more opportunities there will be for future integration with ML and even LLM-based optimization frameworks.
>
>
> Regarding the implementation effort:
> We agree that developing a DP formulation for each new problem requires additional work. However, this is not fundamentally different from what researchers already do in classical SMIP. Even with Benders-type or Lagrangian-based methods, solving NP-hard second-stage subproblems demands significant problem-specific refinement—such as designing tailored cuts, heuristics, or branching strategies.
>
> Moreover, our method can be applied broadly, as many effective exact and heuristic approaches in combinatorial optimization already rely on DP-style operators. Our contribution is to identify suitable DP formulations and accelerate them with GPU parallelism. A well-known analogy is the labeling algorithm widely used in column generation for routing and scheduling problems. Although scenario-level parallelization appears more general, our method exposes both scenario-level and within-DP parallelism, providing a  larger computational boost.
>
> In real-world logistics applications, historical data tend to be inconsistent, and practitioners often retain all available scenarios rather than rely on sampling. The difficulty is that each scenario already represents an NP-hard problem, making even a few dozen scenarios computationally prohibitive. Our GPU-accelerated DP kernels allow us to solve many scenarios simultaneously, enabling the use of much larger scenario sets—something existing methods cannot achieve when the second stage involves integer decisions. As shown in Appendix 5.4, increasing scenario counts is meaningful for improving first-stage decision quality, but previously infeasible at this scale.
>
> For these reasons, we believe the additional implementation effort is justified by the substantial gains in realism, scalability, and solution quality.

---

> > ### Author Response · Authors · 2025-11-21
> > **Response to Questions**
> >
> > > Is there any way to extend this to problems that are not representable as DPs?
> >
> > Great question — and yes, the idea extends well beyond DP. Our framework is not inherently tied to dynamic programming. Any algorithmic component that can be expressed in a matrix-like or tensorized form, or that requires evaluating many scenarios independently, can benefit from the same GPU-based acceleration strategy.
> >
> > We chose to show DP examples here for two reasons:
> > (i) DP is not trivial to parallelize — especially when you want to exploit higher-dimensional parallelism (like across scenarios, transitions, and route options). We wanted to show that even in complex, nested DP structures, you can design kernels that extract massive parallelism — not just across scenarios, but within each scenario’s state transitions. That’s where the real efficiency gain comes from.
> >
> > (ii) DP is wildly used in combinatorial optimization — from exact methods (like labeling algorithms in column generation) to metaheuristics (like split operators in VRP or inventory reinsertion). So by showing how to GPU-accelerate DP, we’re actually accelerating core components of the most widely used algorithms in the field.
> >
> > More broadly, our contribution is a systems-level perspective. Whenever a stochastic optimization model decomposes naturally into a large batch of independent scenario evaluations, which is essentially the defining structure of stochastic programming, our approach applies. The larger goal is to show that stochastic combinatorial optimization does not have to be slow or constrained to small scenario sets. GPU-accelerated evaluation makes it possible to use richer models and much larger scenario collections while retaining exact evaluation.
> >
> >
> > > How many candidate first-stage solutions are used? How does the performance vary with the number of first-stage decisions?
> >
> > As we claim in weakness, our heuristic framework (HGS) operates in a fully end-to-end manner for both case studies. The algorithm continuously generates and improves first-stage candidate solutions, and each candidate is immediately evaluated by computing its total objective value over all scenarios. Therefore, the number of first-stage solutions evaluated is determined organically by the efficiency and runtime of the heuristic rather than being pre-specified. Since HGS is a state-of-the-art metaheuristic for VRP-type problems, it naturally explores a large and diverse set of candidate first-stage decisions; for small instances (for example, those with around ten customers) this effectively covers most of the feasible decision space.
> >
> > In this sense, the variation in the number of first-stage candidates is inherently reflected in the heuristic’s progression: as the search proceeds, more candidates are explored and better first-stage solutions are generated and evaluated. This implicitly captures the relationship between decision quality and the number of first-stage candidates.
> >
> > To make this explicit, we have added two sets of experiments to the revised manuscript and show the results in table here:
> > (1) reporting the number of first-stage candidates evaluated under different settings, and
> > (2) evaluating how solution quality changes with the number of first-stage decisions considered.
> > The results are presented in the **Weakness:Baseline** section, please refer to that.
> > These additional results directly address the reviewer’s question and confirm that the proposed framework behaves consistently across different candidate set sizes.
> >
> > > The word 'training' is somewhat misleading, as it implies a notion of training that is not present. In stochastic programming, it is essentially solving over different-sized scenario sets.
> >
> > Thank you for pointing this out. We used the term “training” to loosely describe the common practice in scenario-based stochastic combinatorial optimization, where we first search for a good first-stage solution using a subset of scenarios (can be one scenario) and then evaluate its performance on a larger set of scenarios. This is analogous to the idea of training and testing in machine learning. However, we agree that this terminology may be misleading, as no actual model training occurs in our method. To avoid confusion, we will revise all instances of “training” and "testing" to more accurately describe this as scenario-based solution search and evaluation.
> >
> >
> > > I believe the authors may be using the ICLR 2025 template.
> >
> > Sorry for that, we have now updated it to 2026 template.
> >
> > > The code is not anonymized.
> >
> > Apologies for the oversight. We have now changed the GitHub link in the introduction, and updated the GitHub README so that all examples can be reproduced.

---

> ### Author Response · Authors · 2025-11-21
> **Response to Weakness part 2**
>
> > **Baselines:**
>
> Thank you for the valuable suggestion. The reason we only do internal comparison is because our goal is not to replace existing heuristic or exact methods, but rather to enable large-scale scenario evaluation within state-of-the-art heuristics.
>
> While the proposed matrix-based DP operators can, in principle, be integrated into any metaheuristic framework, we chose HGS due to its flexibility, effectiveness, and consistently strong empirical performance across diverse VRP variants [4,5]. In addition to its extensive use in prior research, HGS has demonstrated top-tier performance in recent benchmarking competitions, including the DIMACS VRP Challenge and the EURO/NeurIPS Vehicle Routing Competition [6], where it formed the core of leading solution methods. These results further support the robustness of HGS as a general-purpose framework for complex routing problems.
>
> We agree that comparisons against exact methods would be informative, even if limited to smaller instances. However, we are not aware of existing work that develops a Benders or L-shape method for this specific problem structure, and implementing such a method from scratch is beyond the scope of this paper. Instead, we will include a comparison with a standard solver (Gurobi) on the extensive form in the final version. The model can be found in the Appendix of the updated manuscript.
>
> |  |  | #Scenario | 100 | 500 | 1000 | 2000 | 10000 |
> |-|-|-|-|-|-|-|-|
> |  | Time (s.) | GPUSPLIT | 13.76 | 28.92 | 52.64 | 95.42 | 420.11 |
> | 6 Customer |  | MILP | 26.7 | 587.33 | 1708.51 | **3600** | **OOM** |
> |  | Obj. Value | GPUSPLIT | 3399.59 | 3392.72 | 3391.86 | 3391.43 | 3391.17 |
> |  |  | MILP | 3399.59 | 3392.72 | 3391.86 | **10541.3** | **-** |
> |  | Time (s.) | GPUSPLIT | 485.03 | 3215.46 | **3600** | **3600** | **3600** |
> | 10 Customer |  | MILP | **3600** | **3600** | **3600** | **3600** | OOM |
> |  | Obj. Value | GPUSPLIT | 3797.37 | 3804.89 | 3800.62 | 3802.95 | 3955.16 |
> |  |  | MILP | 3797.37 | 3804.89 | **3913.01** | **5336.41** | **-** |
>
> We use the state-of-the-art MILP solver Gurobi to solve the CVRPSD extensive form under varying numbers of scenarios and customers, and compare its performance with our GPU-based splitting algorithm. The results show a clear scalability gap: while GPUSPLIT’s runtime grows roughly linearly with the number of scenarios, the CPU-based MILP quickly becomes impractical, hitting time limits and eventually exhausting memory.
> In addition, for larger scenario sets, GPUSPLIT remains tractable and often achieves better objective values. For example, on the 10-customer dataset with over 1,000 scenarios, neither method proves optimality, but GPUSPLIT consistently finds higher-quality solutions than the CPU-based solver.
>
> Regarding DSIRP, we provide a comparison with the state-of-the-art method in [3], which solves DSIRP under a single scenario.
>
> | Instance | [3] Obj. | [3] Time (s) | Ours Obj. | Ours Time (s) | Gain (%) |
> |-|-|-|-|--|-|
> | small (n < 50)           | 9131.50      | 67.10        | 8703.20| 102.30| 4.68|
> | medium (50 \leq n \leq 100) | 30137.80   | 888.30       | 28219.00| 1029.80| 6.37|
> | large (n > 100)          | 60051.40     | 9248.90      | 57523.30| 7200.00| 4.21|
> | Average                      | 33106.87     | 3401.40      | 31481.83| 2777.40| 5.09|
>
> Across instances of various sizes, our approach achieves 4.21–6.37% lower objective values (5.09% on average), while keeping runtimes in the same order of magnitude and even running faster on the largest class. This demonstrates that explicitly handling multiple demand scenarios can yield consistently higher-quality DSIRP solutions without incurring prohibitive additional computation compared to [3].

---

> ### Author Response · Authors · 2025-11-21
> **Response to Weakness part 3**
>
> > **Clarity**
>
> We thank the reviewer for pointing this out. We realize that some aspects of the presentation may have led to confusion. Indeed, our high-level heuristic approach (HGS) works in an end-to-end manner for both case studies. Conceptually, the outer loop of HGS continuously generates promising first-stage candidates, which are then evaluated under a large number of scenarios using our GPU-based evaluation procedure. In this sense, the first-stage decisions are “received” naturally as part of the heuristic search, and the search itself is guided toward improving solution quality.
>
> Regarding the number of candidates evaluated, we have added experimental results to address this question. The quality of first-stage decisions over a varied number of candidates is an important consideration. In fact, Figure 6 already illustrates this indirectly: as the heuristic runs longer, it discovers better first-stage solutions, which can be interpreted as evaluating a larger and more diverse set of candidates. To make this point more explicit, we have now added an additional set of experiments on the DSIRP instance, which directly reports the evaluation of different first-stage solutions and their corresponding performance under large scenario sets.
>
> | Time (s.) | GPU Calls | # Candidates | Obj. | Gain |
> |------------|-----------|------------------------|-----------|----------------------|
> | 120 | 1,380 | **92** | 77,898,383 | - |
> | 240 | 2,715 | **181** | 77,840,004 | 0.07% ↓ |
> | 360 | 4,065 | **271** | 77,804,713 | 0.12% ↓ |
> | 480 | 5,415 | **361** | 77,741,966 | 0.20% ↓ |
> | 600 | 6,750 | **450** | 77,741,966 | 0.20% ↓ |
>
> The candidate first-stage decisions are generated by a genetic algorithm and evaluated via GPU, where each GPU call solves a 500-scenario DP subproblem in parallel. On the 15-customer, 500-scenario instance, this results in 92–450 evaluated candidates depending on the time budget, during which the framework consistently finds equal or better solutions as more candidates are considered. The reported results clearly show how solution quality evolves with the number of evaluated candidates, thereby directly illustrating the behavior and viability of our approach under varying candidate counts.
>
> | Time (s.) | GPU Calls | # Candidates | Obj. | Gain |
> |---|---|---|---|---|
> | 120 | 304,740 | 304,740 | 3,893.32 | - |
> | 240 | 611,841 | 611,841 | 3,811 | 2.11%↓ |
> | 360 | 914,889 | 914,889 | 3,800.62 | 2.38%↓ |
> | 480 | 1,216,803 | 1,216,803 | 3,800.62 | 2.38%↓ |
> | 600 | 1,524,407 | 1,524,407 | 3,800.62 | 2.38%↓ |
>
> We conducted the same set of experiments for CVRPSD and observe the same behavior as in DSIRP: as the time budget increases and more first-stage candidates are explored, the framework consistently returns equal or better solutions, with no degradation in solution quality. The experimental results clearly track how the objective improves with the number of evaluated candidates, further confirming the robustness and viability of our candidate-based GPU evaluation scheme.

---

> ### Author Response · Authors · 2025-11-21
> **Response to Weakness part 4**
>
> We fully agree that our work does not involve training ML models and instead focuses on exact evaluations. This indeed makes OR-focused venues a natural fit. However, we also believe that the ML community has a growing interest in GPU-accelerated algorithmic primitives, particularly through work on algorithm unrolling and differentiable optimization. These lines of research show that ML venues increasingly value contributions that expose efficient computational structures, even when no learning component is present
> There is also a well-developed line of work on algorithm unrolling [1,2], in which neural networks are constructed by unrolling the iterations of existing non-ML algorithms. This paradigm effectively connects classical general-purpose algorithms with GPU-oriented learning models. Our work can be viewed as a broader reinterpretation of classical dynamic programming schemes, which may in turn motivate new developments such as DP-based unrolled architectures.
>
> On the other hand, from my experience of over eight years working in VRP and stochastic programming within traditional OR communities, I have noticed that research emphasizing large-scale computational acceleration and implementation-oriented innovations (e.g., GPU-based acceleration or even deep learning methods for VRP family), which are less theory-driven but technically challenging, often receives limited recognition despite its practical significance.
>
> Our motivation for submitting to ICLR is to bridge this gap and to invite broader attention from the ML community, where GPU-based computation has been transformative, and in turn attract more OR researchers to explore this promising direction. We see this work as an initial step to highlight how GPU architectures can empower large-scale stochastic combinatorial optimization. That said, this is only one direction we have explored for leveraging GPU’s matrix capabilities, and we are well aware that our methods are still not generic enough. Our hope is to spark broader interest and encourage more OR researchers to develop GPU-based approaches from different perspectives. While our approach is not ML-based, we believe it aligns with ICLR’s growing interest in the intersection of optimization, stochasticity, and computation, and we hope it can inspire future research that unites these perspectives.
>
> [1] Zhang, Jian, and Bernard Ghanem. "ISTA-Net: Interpretable optimization-inspired deep network for image compressive sensing." Proceedings of the IEEE conference on computer vision and pattern recognition. 2018.
>
> [2] Li, Bingheng, et al. "Pdhg-unrolled learning-to-optimize method for large-scale linear programming." arXiv preprint arXiv:2406.01908 (2024).
>
> [3] Coelho, Leandro Callegari, Gilbert Laporte, and Jean-François Cordeau. Dynamic and stochastic inventory-routing. Montreal: CIRRELT, 2012.
>
> [4] Vidal, Thibaut, et al. "A hybrid genetic algorithm with adaptive diversity management for a large class of vehicle routing problems with time-windows." Computers & operations research 40.1 (2013): 475-489.
>
> [5] Vidal, Thibaut, et al. "A unified solution framework for multi-attribute vehicle routing problems." European Journal of Operational Research 234.3 (2014): 658-673.
>
> [6] Kool, Wouter, et al. "The EURO meets NeurIPS 2022 vehicle routing competition." NeurIPS 2022 Competition Track. PMLR, 2023.

---

> ### Author Response · Authors · 2025-11-26
>
> Dear Reviewer 7hTf,
> We sincerely appreciate your thoughtful and constructive feedback. We hope that our responses have satisfactorily addressed your concerns. If you have any further questions or comments, please do not hesitate to let us know. We look forward to hearing from you.

---

> > ### Comment · Reviewer_7hTf · 2025-11-27
> >
> > Thanks a lot for the detailed clarification.  The authors have sufficient addressed a lot of my concerns, and I have raised my score accordingly.
> >
> > Overall, the concerns I still have are with respect to limitations in scope (DP second-stage) and as novelty with "batched min–plus matrix–vector” as other reviewers have pointed out.  In addition, I am on the fence regarding the venue fit still.  However, I do agree GPU-based numerical algorithms may be of interest to the broader ML community.  Despite these weaknesses, the authors present a highly scalable approach for SP, which has gained some interest in recent years by the ML community, so acceptance is worth consideration.

---

> > > ### Author Response · Authors · 2025-11-28
> > >
> > > We appreciate the reviewer’s positive assessment and the constructive feedback. Regarding the remaining concern, we would like to emphasize that, building on the comments from other reviewers, we have now clearly distinguished our work from generic batched min–plus matrix–vector operations. In the revision, we explicitly highlight the aspects that go beyond existing formulations: the hardware-aware kernel design tailored to DP structure, the numerically safe state/action masking, and the multi-axis parallelism across scenarios, DP layers, and transition options. These elements collectively allow full Bellman updates for exact integer second-stage models at scales exceeding 10⁶ scenarios, which, to our knowledge, has not been demonstrated by prior GPU-based approaches.
> > > Once again, we sincerely appreciate the reviewer’s thorough reading of the paper and the time invested in evaluating both the original submission and our revision. Your feedback has helped us significantly improve the clarity and positioning of the work.

---

### Author Response · Authors · 2025-11-29
**Letter to AC and all Reviewers**

We thank all reviewers for the time and effort they invested in reading the paper and engaging in detailed discussion. **For the ease of understanding what was covered during the rebuttal**, we briefly summarize the key clarifications, revisions, and additional experiments provided in response. We have taken all reviewer comments, both positive and critical, very seriously and have substantially revised the paper accordingly.

# Discussion with Reviewer 7hTf

We sincerely thank the reviewer for the detailed and constructive engagement. Their initial concerns focused on
#### (i) the DP-structured scope,
#### (ii) lack of external baselines and unclear decision quality,
#### (iii) how first-stage candidates are generated/evaluated,
#### (iv) venue fit and terminology.

During the rebuttal we clarified that the contribution does not lie in the min--plus formulation but in the hardware-aware, scenario-batched GPU kernels that enable exact DP-based second-stage models at scale. We added extensive-form MILP baselines (CVRPSD) and a comparison with the state-of-the-art DSIRP method, both showing competitive or improved decision quality while scaling to scenario counts classical methods cannot handle.

We also added experiments tracking the number of evaluated first-stage candidates and the resulting objective improvements, addressing the reviewer’s concerns about viability and clarity. Template and code anonymization issues were fixed.

The reviewer confirmed that these clarifications “sufficiently addressed” their concerns, raised their score (from 2 to 6), and stated that acceptance is worth consideration despite remaining reservations about scope and venue fit.

# Discussion with Reviewer df1c

We thank the reviewer for the careful assessment and constructive suggestions. Their concerns focused on (i) the need for end-to-end comparisons rather than operator-level timing, (ii) missing external benchmarks for solution quality, and (iii) the absence of a dedicated Related Work section.

During the rebuttal, we clarified that our experiments already integrate the GPU operator within a full solving framework (HGS). To address baseline concerns, we added extensive-form MILP comparisons (CVRPSD) and a comparison with the state-of-the-art DSIRP method, both showing that our approach produces competitive or improved solutions while scaling to scenario counts that exact methods cannot handle. We also added CPU vs.\ GPU tests using the same matrix-based DP formulation to isolate hardware effects. A full Related Work section was added to the Appendix as requested.

# Discussion with Reviewer 8Uwb

We thank the reviewer for the careful reading and detailed technical feedback. The reviewer’s expertise in GPU-related methods has clearly strengthened our work. Their concerns focused on
#### (i) overstated novelty and missing citations to tropical/semiring DP and prior GPU-DP work
#### (ii) mixing hardware and algorithmic effects in CPU--GPU comparisons
#### (iii) lack of quantitative memory analysis
#### (iv) limited scope beyond the two DP case studies.

During the rebuttal, we substantially revised the paper. We added explicit acknowledgements of prior work and reframed the contribution around the scenario-batched, multidimensional GPU kernels rather than the min–plus formulation itself. We added CPU baselines implementing the same matrix-based DP algorithm to isolate hardware effects, as well as extensive-form MILP and state-of-the-art DSIRP comparisons showing competitive or improved decision quality. We also included detailed GPU memory measurements (up to one million
scenarios) and discussion of the scenario--state-size trade-off. Finally, we clarified how our approach differs from earlier GPU-DP methods and why it serves as a systems-level contribution for large-scale SP.

Overall, we believe these additions fully address the reviewer’s technical questions, and we expect their score to improve accordingly once they review the new material.

# Discussion with Reviewer fYpR

We thank the reviewer for a very positive and encouraging assessment. They viewed the contribution as novel and impactful, highlighting that our GPU-based DP reformulation enables millions of scenarios and leads to improved SAA solution quality under realistic time budgets, with clear toy examples and solid experiments.

Their concerns were mostly minor: (i) limitations of applicability beyond the discrete DP setting, (ii) behavior under irregular state spaces, (iii) the desire for a unified algorithm in the main text, (iv) unclear use of “DAG”, and (v) notation and terminology inconsistencies. During the rebuttal, we clarified the discrete-state/action assumption and its implications for continuous problems, explained how masking/padding handles irregular state spaces (and the associated efficiency trade-offs), added a unified high-level GPU-DP algorithm in the main body, removed unnecessary DAG terminology, and cleaned up notation and typos.

---

> ### Author Response · Authors · 2025-11-29
> **Summary**
>
> Our work originates from real logistics applications where companies must evaluate large sets of historical scenarios because the data are highly inconsistent and cannot be reliably modeled or sampled. Each scenario itself contains an NP-hard second-stage problem, making even a few dozen scenarios extremely expensive to solve with existing SMIP methods. The practical need, to evaluate hundreds or thousands of full-fidelity scenarios in reasonable time, is what drove the development of this framework. This paper is not about designing a clever DP for its own sake; it is about solving a real computational bottleneck that practitioners repeatedly face.
>
> To our knowledge, no existing SMIP approach, whether Benders, Lagrangian, cutting-plane, or ML-based approximation, can handle NP-hard second-stage problems at this scenario scale. This limitation is precisely why most SP papers reduce the second stage to LPs or simplified variants. Our contribution is to make exact integer second-stage solves tractable at scenario counts that were out of reach before, without sacrificing model fidelity. Achieving this through GPU-accelerated DP kernels is, in our view, a meaningful step forward for SMIP and fills a long-standing computational gap.
>
> Although “DP” may sound specialized, it underlies many core optimization algorithms. DP is not a niche technique. Many widely used exact and heuristic combinatorial optimization methods, including labeling algorithms in column generation for routing and scheduling, are DP-style operators. Our contribution is to accelerate these operators so they can be executed across thousands or millions of scenarios in parallel. This substantially broadens their applicability and practical impact.
>
> GPUs fundamentally change what is computationally feasible in scenario-based optimization. Our results show that GPU architectures can shift the boundary of what SMIP researchers consider tractable. For problems with structured transition costs, common in routing, scheduling, and inventory, the combination of scenario-level and intra-DP parallelism yields orders-of-magnitude gains. This makes large-scale recourse modeling a realistic option rather than an academic ideal.
>
> This direction opens new opportunities for both OR and ML communities.
> By making full-fidelity recourse problems tractable, the framework expands the space in which OR and ML can meaningfully interact. This includes:
> ML researchers working on differentiable optimization and SPO methods
> policy-learning or RL models that require fast exact subproblem solves
> LLM-based decision systems that rely on structured optimization or recourse evaluation
> OR researchers developing new SMIP, decomposition, or DP formulations
> We view this work as an early step toward a broader research area: GPU-accelerated structured decision optimization. Our hope is that this paper encourages researchers from both communities to explore this direction further.

---

### Meta-Review · Area_Chair_AkYp · 2026-01-14

**Summary:**

This paper demonstrates how the sequential structure of dynamic programming methods can be reformulated into min--plus matrix--vector products computations that can directly exploit GPU kernels, leading to 2D and 3D parallelized applications.  Neat formulation,  convincing experiments and open source CUDA contributions were highlighted as strengths. Most weaknesses appear to have been carefully addressed in comprehensive rebuttal responses.

**Reviewer Concerns:**

Reviewer 7hTf:  Concerns on applicability to Dynamic programming methods in ML and ICLR fit; baselines and clarity were largely addressed leading the reviewer to switch their initial assessment of the paper.

Reviewer df1c: Concerns on benchmarking rigor, weak baselines, and weak related works. The authors responded with extensive comparisons showing competitive or improved solutions while scaling to scenario counts that exact methods cannot handle. They also updated the related works section.

Reviewer 8Uwb: Concerns about overstated novelty, CPU-GPU mixup, lack of memory analysis, limited scope. Most of these appear to have been addressed.

Reviewer fYpR: mostly minor concerns.

**Reviewer Scores:**

Reviewer 7hTf: raised their score (from 2) and recommended acceptance; this is not reflected in the current dashboard.
Reviewer df1c: would likely have raised their score.
Reviewer 8Uwb: may have raised their score, not sure.
Reviewer fYpR: already at 6, unlikely to have changed.

---

### Decision · Program_Chairs · 2026-01-26

Accept (Poster)